# Simple Convergence Proof of Adam From a Sign-Like Descent Perspective

## Abstract

Adam is widely recognized as one of the most effective optimizers for training deep neural networks (DNNs). Despite its remarkable empirical success, its theoretical convergence analysis remains unsatisfactory. Existing works predominantly interpret Adam as a preconditioned stochastic gradient descent with momentum (SGDM), formulated as $\boldsymbol{x}_{t+1} = \boldsymbol{x}_t - \frac{\gamma}{\sqrt{\boldsymbol{v}_t} + \epsilon} \circ \boldsymbol{m}_t$. This perspective necessitates strong assumptions and intricate techniques, resulting in lengthy and opaque convergence proofs that are difficult to verify and extend. While many prior works have treated Adam as a sign-like optimizer to interpret its practical advantages, we are the first to formally provide a convergence proof for Adam from the perspective of sign-like descent, expressed as $\boldsymbol{x}_{t+1} = \boldsymbol{x}_t - \gamma \frac{|\boldsymbol{m}_t|}{\sqrt{\boldsymbol{v}_t} + \epsilon} \circ \mathrm{Sign}(\boldsymbol{m}_t)$. This reformulation significantly simplifies the convergence analysis. For the first time, with some mild conditions, we prove that Adam achieves the optimal rate of $\mathcal{O}(\frac{1}{T^{1/4}})$ rather than the previous $\mathcal{O}\left(\frac{\ln T}{T^{1/4}}\right)$ under weak assumptions of the generalized $p$-affine variance and $(L_0, L_1, q)$-smoothness, without dependence on the model dimensionality or the numerical stability parameter $\epsilon$. Additionally, our theoretical analysis provides new insights into the role of momentum as a key factor ensuring convergence and offers practical guidelines for tuning learning rates in Adam, further bridging the gap between theory and practice.

## 1 Introduction

Currently, Adam (Kingma & Ba, 2015) has emerged as the predominant optimizer for training Transformers (Vaswani et al., 2017), particularly for state-of-the-art large language models (LLMs) (Brown et al., 2020; Chowdhery et al., 2023; Touvron et al., 2023a) and large vision models (Radford et al., 2021; Kirillov et al., 2023). Notably, Adam's influence extends beyond Transformers to modern convolutional neural networks (CNNs), such as ConvNeXt (Liu et al., 2022; Woo et al., 2023), where it has become the de facto choice for optimization. This is despite the traditional preference for stochastic gradient descent (SGD) (Krizhevsky et al., 2017; He et al., 2016), which was historically considered more suitable for CNN training.

However, the theoretical convergence analysis of Adam lags behind its significant practical success. The original proof in (Kingma & Ba, 2015) was based on the convexity of the objective function but was later found to be flawed (Reddi et al., 2018). To address this, AMSGrad, a fixed variant of Adam, was proposed, but its theoretical analysis still relied on the convexity assumption. Chen et al. (2018) were the first to theoretically demonstrate that a class of Adam-type optimizers, including AMSGrad and AdaFom, converge to stationary solutions for non-convex problems. Subsequently, Défossez et al. (2020) provided a simplified proof analyzing the convergence rates of vanilla Adam and Adagrad. However, their analysis required $\beta_1 < \beta_2$ and depended on the model dimensionality $d$. Chen et al. (2022) introduced practical, easy-to-check conditions to ensure the global convergence of Adam, but their proved convergence rate also heavily

Table 1: Comparison of different convergence proofs for Adam. "FCT" refers to the full corrective term. "Conv. Rate" denotes the convergence rate for approaching stationary points (*i.e.*, $\|\nabla F(\boldsymbol{x}_T)\|_2 \to 0$). $T$ represents the number of iterations, $E$ the number of epochs, $d$ the model dimensionality, $n$ the total number of samples, and $\epsilon$ the numerical stability parameter.

| References | FCT | Noise Condition | Smooth Condition | Coeff. Condition | Conv. Rate |
|---|---|---|---|---|---|
| (Chen et al., 2018) | No | Bounded Grad. | $L$-Smooth | $\beta_{1_t} \leq \beta_1,$ $\beta_{2_t} = 1 - \frac{1}{t}$ | $\mathcal{O}\left(\frac{d^{1/2}\epsilon^{-1}\ln T}{T^{1/4}}\right)$ |
| (Défossez et al., 2020) | No | Bounded Grad. | $L$-Smooth | $\beta_2 < \beta_1,$ $\beta_2 = 1 - \frac{1}{T}$ | $\mathcal{O}\left(\frac{d^{1/2}\ln(\epsilon^{-1}T)}{T^{1/4}}\right)$ |
| (Chen et al., 2022) | No | Bounded Grad. | $L$-Smooth | $\boldsymbol{\beta_{2_t} < \sqrt{\beta_1}},$ $\beta_{2_t} = 1 - \frac{1}{t}$ | $\mathcal{O}\left(\frac{d^{3/4}\ln\epsilon^{-1}\ln T}{T^{1/4}}\right)$ |
| (Zhang et al., 2022) | No | Affine Var. | $L$-Smooth | $\boldsymbol{\beta_2 < \sqrt{\beta_1}},$ $\beta_2 = 1 - \mathcal{O}(\frac{1}{n^3})$ | $\mathcal{O}\left(\frac{n^{1/2}d^{3/4}\ln E}{E^{1/4}}\right)$ |
| (Wang et al., 2023c) | No | Bounded Var. | $(L_0, L_1)$-Smooth | $\boldsymbol{\beta_2 < \sqrt{\beta_1}},$ $\beta_2 = 1 - \mathcal{O}(\frac{1}{T})$ | $\mathcal{O}\left(\frac{n^{1/2}d^{1/2}\ln E}{E^{1/4}}\right)$ |
| (Li et al., 2023) | Yes | sub-Gaussian Var. | **Generalized** $\boldsymbol{(L_0, L_1, q)}$-**Smooth** | $\boldsymbol{\beta_2 = 1 - \mathcal{O}(\frac{1}{T^{1/2}})}$ | $\mathcal{O}\left(\frac{\epsilon^{-2}\ln T}{T^{1/4}}\right)$ |
| (Hong & Lin, 2025) | Yes | Affine Var. | **Generalized** $\boldsymbol{(L_0, L_1, q)}$-**Smooth** | $\beta_2 < \beta_1,$ $\beta_2 = 1 - \mathcal{O}(\frac{1}{T})$ | $\mathcal{O}\left(\frac{d\ln(\epsilon^{-1}T)}{T^{1/4}}\right)$ |
| **Corollary 3** [1] | Yes | **Generalized** $p$-**Affine Var.** | **Generalized** $\boldsymbol{(L_0, L_1, q)}$-**Smooth** | $\boldsymbol{\beta_2 < \sqrt{\beta_1}},$ $\boldsymbol{\beta_2 = 1 - \mathcal{O}(\frac{1}{T^{3/4}})}$ | $\boldsymbol{\mathcal{O}\left(\frac{1}{T^{1/4}}\right)}$ |

1. Compared to previous works, in Theorem 3.1 we establish the convergence of vanilla Adam under the weaker assumptions of generalized $p$-affine variance and $(L_0, L_1, q)$-smoothness (see Section 2 for definitions). Furthermore, we are the first to prove that Adam achieves the optimal convergence rate of $O(\frac{1}{T^{1/4}})$ in a dimension-free and $\epsilon$-independent manner, improving upon the previous rate of $\mathcal{O}\left(\ln T/T^{1/4}\right)$. Note that our primary convergence result (Theorem 3.1) requires Conditions 1-3, while Theorem B.6 in the appendix provides a weaker result without them.

relied on the model dimensionality $d$. Notably, the analyses in (Chen et al., 2018; Défossez et al., 2020; Chen et al., 2022) all assumed bounded stochastic gradients. Later, Zhang et al. (2022) provided a theoretical proof for random-reshuffling Adam under the weaker affine variance assumption. However, this proof achieved a slower convergence rate with an epoch-complexity bound and relies on the total number of samples. Additionally, these works assumed the conventional uniformly bounded smoothness condition, *i.e.*, the $L$-smoothness condition. Recent studies, however, have shown that the $L$-smoothness assumption is inadequate for optimizing complex DNNs such as LSTMs and Transformers (Zhang et al., 2019; Crawshaw et al., 2022). Instead, it should be relaxed to the non-uniform $(L_0, L_1)$-smoothness condition (see Section 2 for details). Recently, Wang et al. (2023c) analyzed random-reshuffling Adam under the $(L_0, L_1)$-smoothness assumption, but their theoretical convergence rate was still based on epoch complexity and depended on the total number of samples. Li et al. (2023) demonstrated that Adam provably converges to stationary points with the optimal rate under generalized $(L_0, L_1, q)$-smoothness. However, this bound heavily relied on a large $\epsilon$, making Adam behave similarly to SGD and losing its adaptive properties. Most recently, Hong & Lin (2025) established the convergence rate of a simplified Adam under both the affine variance and the generalized $(L_0, L_1, q)$-smoothness assumptions. However, their results still heavily depended on the model dimensionality. A detailed comparison of these convergence analyses for Adam is provided in Table 1.

All existing theoretical convergence proofs for Adam are path-dependent, treating Adam as a preconditioned SGD with momentum, as initially described in (Kingma & Ba, 2015), *i.e.*, $\boldsymbol{x}_{t+1} = \boldsymbol{x}_t - \frac{\gamma}{\sqrt{\boldsymbol{v}_t}+\epsilon} \circ \boldsymbol{m}_t$ where $\sqrt{\boldsymbol{v}_t} + \epsilon$ serves to precondition $\boldsymbol{m}_t$, introducing an effective learning rate of $\frac{\gamma}{\sqrt{\boldsymbol{v}_t}+\epsilon}$. This preconditioned formulation not only requires strong assumptions and intricate techniques for theoretical convergence analysis but also leads to proofs that are complex, lengthy, and difficult to verify or extend. Additionally, such theoretical analyses provide limited insights for practical optimization with Adam or for further enhancing the algorithm.

On the other hand, recent empirical evidence suggests that Adam's effectiveness may primarily stem from its sign-like property Balles & Hennig (2018). Kunstner et al. (2023) empirically demonstrates that sign descent with momentum achieves performance comparable to Adam when training Transformers, albeit without comprehensive analytical justification. Similarly, Chen et al. (2023b) employs an AutoML approach to discover a highly effective optimizer, Lion, which resembles signSGD with momentum and outperforms Adam across various DNN models. Kunstner et al. (2024) observed that Adam's superior performance on language models can be attributed to its sign-like property, which is particularly advantageous in addressing heavy-tailed class imbalance. Recently, Muon Jordan et al. (2024), an extended matrix-sign optimizer, has demonstrated significant potential for training DNNs Liu et al. (2025); Shah et al. (2025). However, no existing theoretical convergence proof for Adam considers its resemblance to sign descent, leaving its efficacy unexplained.

To address the aforementioned issues, we treat Adam as a stochastic sign-like descent optimizer to analyze its convergence. Specifically, we reformulate Adam as: $\boldsymbol{x}_{t+1} = \boldsymbol{x}_t - \gamma \frac{|\boldsymbol{m}_t|}{\sqrt{\boldsymbol{v}_t}+\epsilon} \circ \text{Sign}(\boldsymbol{m}_t)$ where we take $\frac{|\boldsymbol{m}_t|}{\sqrt{\boldsymbol{v}_t}+\epsilon}$ as a single random variable. This reformulation not only completely circumvents the aforementioned challenges but also simplifies the proof process. Moreover, the provable convergence rate of the gradient norm in expectation achieves the optimal rate under the weak assumptions of non-uniform smoothness and affine variance noise without dependency on the model dimensionality $d$ and the numerical-stability parameter $\epsilon$. Additionally, this theoretical analysis enhances our understanding of the foundations underlying Adam's success. It sheds light on why momentum improves convergence, and how to better tune hyperparameters.

Our contributions are summarized as follow:

- We pioneer the establishment of a theoretical convergence proof for vanilla Adam from the perspective of sign-like descent. This approach circumvents the intractable challenges of preconditioned settings and significantly simplifies the proof process.

- We are the first to prove that vanilla Adam achieves the convergence rate of $\mathcal{O}(\frac{1}{T^{1/4}})$, compared to the previous $\mathcal{O}\left(\frac{\ln T}{T^{1/4}}\right)$, under the weak assumptions of generalized $p$-affine noise and $(L_0, L_1, q)$-smoothness along with some mild condidtions, without reliance on the model dimensionality or the numerical stability parameter $\epsilon$.

- Our theoretical convergence analysis provides the insight into the significance of momentum and provides guidance on tuning the learning rate in Adam.

## 2 PRELIMINARY

### 2.1 NOTATION

In this paper, the optimizer aims to minimize the empirical risk loss of a model on a dataset, *i.e.*,

$$\min_{\boldsymbol{x}\in\mathbb{R}^d} F(\boldsymbol{x}) = \mathbb{E}_{\zeta\sim\mathcal{D}}[f(\boldsymbol{x};\zeta)] = \frac{1}{n}\sum_{i=1}^{n} f(\boldsymbol{x};\omega_i), \tag{1}$$

where $\boldsymbol{x} \in \mathbb{R}^d$ and $\zeta$ are independently and identically sampled from the dataset $\{\omega_i : \omega_i \in \mathcal{D}, 1 \leq i \leq n\}$. For simplicity, we sometimes use $\boldsymbol{g} = \nabla f(\boldsymbol{x};\zeta)$.

As shown in Table 1, the previous studies (Chen et al., 2018; Défossez et al., 2020; Zhang et al., 2022; Wang et al., 2023c) have omitted the bias correction in Line 6-7 of Algorithm 1 when analyzing the convergence rate, while we will provide the theoretical analysis of vanilla Adam with bias correction.

---

**Algorithm 1.** Adam

---

1: **Input**: the momentum $\boldsymbol{m}_0 = 0$, $\boldsymbol{v}_0 = 0$, the numerical stable constant $\epsilon$, the exponential moving average coefficient $\beta_1$ and $\beta_2$, and the learning rate $\gamma$.

2: **for** $t = 1, ..., T$ **do**

3:     Randomly sample data and compute the gradient: $\boldsymbol{g}_t \leftarrow \nabla f(\boldsymbol{x}_t; \zeta_t)$

4:     Update the momentum $\boldsymbol{m}_t$: $\boldsymbol{m}_t \leftarrow \beta_1 \boldsymbol{m}_{t-1} + (1 - \beta_1) \boldsymbol{g}_t$

5:     Update the momentum $\boldsymbol{v}$: $\boldsymbol{v}_t \leftarrow \beta_2 \boldsymbol{v}_{t-1} + (1 - \beta_2) \boldsymbol{g}_t^2$

6:     Compute the bias corrected $\hat{\boldsymbol{m}}_t$: $\hat{\boldsymbol{m}}_t \leftarrow \frac{\boldsymbol{m}_t}{1 - \beta_1^t}$

7:     Compute the bias corrected $\hat{\boldsymbol{v}}_t$: $\hat{\boldsymbol{v}}_t \leftarrow \frac{\boldsymbol{v}_t}{1 - \beta_2^t}$

8:     Update the model parameter: $\boldsymbol{x}_{t+1} \leftarrow \boldsymbol{x}_t - \gamma \frac{\hat{\boldsymbol{m}}_t}{\sqrt{\hat{\boldsymbol{v}}_t} + \epsilon}$

9: **end for**

---

## 2.2 DETAILS OF ADAM

To facilitate the analysis of Adam, we provide the details of Adam in Algorithm 1.

## 2.3 ASSUMPTIONS AND CONDITIONS

To analyze the convergence rate of Adam, we list the main assumption as follows.

**Assumption A** [Bounded Infimum]. *There exists a constant $F^* > -\infty$, and the objective function follows $F(\boldsymbol{x}) \geq F^*$ for any $\boldsymbol{x} \in \mathbb{R}^d$.*

**Assumption B.1** [$L$-Smoothness] *There exists a constants $L \geq 0$, and then for any $\boldsymbol{x}, \boldsymbol{y} \in \mathbb{R}^d$, the gradient of the objective function follows*

$$\|\nabla F(\boldsymbol{y}) - \nabla F(\boldsymbol{x})\|_2 \leq L\|\boldsymbol{x} - \boldsymbol{y}\|_2. \tag{2}$$

**Assumption B.2** [$(L_0, L_1)$-Smoothness] *There exist constants $L_0, L_1 \geq 0$, and then for any $\boldsymbol{x}, \boldsymbol{y} \in \mathbb{R}^d$, the gradient of the objective function follows*

$$\|\nabla F(\boldsymbol{y}) - \nabla F(\boldsymbol{x})\|_2 \leq (L_0 + L_1\|\nabla F(\boldsymbol{x})\|_2)\|\boldsymbol{x} - \boldsymbol{y}\|_2. \tag{3}$$

**Assumption B.3** [$(L_0, L_1, q)$-Smoothness] *There exist constants $L_0, L_1 > 0$ and $0 \leq q \leq 1$, and then for any $\boldsymbol{x}, \boldsymbol{y} \in \mathbb{R}^d$, the gradient of the objective function follows*

$$\|\nabla F(\boldsymbol{y}) - \nabla F(\boldsymbol{x})\|_2 \leq (L_0 + L_1\|\nabla F(\boldsymbol{x})\|_2^q)\|\boldsymbol{x} - \boldsymbol{y}\|_2. \tag{4}$$

When $q = 1$, the generalized $(L_0, L_1, q)$-smoothness (Assumption B.3) is reduced to the $(L_0, L_1)$-smoothness (Assumption B.2). When $L_1 = 0$ or $q = 0$, the generalized $(L_0, L_1, q)$-smoothness (Assumption B.3) is reduced to the standard $L$-smoothness (Assumption B.1). $(L_0, L_1)$-smoothness was originally defined in (Zhang et al., 2019) as a bound on the second-order Hessian function. Following Zhang et al. (2020), we reformulate the $(L_0, L_1)$-smoothness as an affine form of the gradient norm for first-order differentiable functions. Subsequently, Li et al. (2023) first introduced the generalized $(L_0, L_1, q)$-smoothness to analyze the convergence of Adam, followed by Wang et al. (2024) and Hong & Lin (2025).

**Assumption C.1** [Bounded Variance]. *There exists a positive constant $\sigma_0 > 0$, and then for any $\boldsymbol{x}_t \in \mathbb{R}^d$, the noisy gradient of the objective function obeys*

$$\mathbb{E}[\nabla f(\boldsymbol{x}; \zeta)] = \nabla F(\boldsymbol{x}), \quad \mathbb{E}[\|\nabla f(\boldsymbol{x}; \zeta) - \nabla F(\boldsymbol{x})\|_2^2] \leq \sigma_0^2. \tag{5}$$

**Assumption C.2** [Affine Variance]. *There exist constants $\sigma_0, \sigma_1 \geq 0$, and then for $\boldsymbol{x} \in \mathbb{R}^d$, the noisy gradient of the objective function obeys*

$$\mathbb{E}[\nabla f(\boldsymbol{x}; \zeta_t)] = \nabla F(\boldsymbol{x}), \quad \mathbb{E}[\|\nabla f(\boldsymbol{x}; \zeta) - \nabla F(\boldsymbol{x})\|_2^2] \leq \sigma_0^2 + \sigma_1^2\|\nabla F(\boldsymbol{x})\|_2^2. \tag{6}$$

**Assumption C.3** [ $p$-Affine Variance]. *There exist constants $\sigma_0, \sigma_1 \geq 0$ and $0 \leq p \leq 2$, and then $\boldsymbol{x} \in \mathbb{R}^d$ at any time, the noisy gradient of the objective function obeys*

$$\mathbb{E}[\nabla f(\boldsymbol{x}; \zeta_t)] = \nabla F(\boldsymbol{x}), \qquad \mathbb{E}[\|\nabla f(\boldsymbol{x}; \zeta) - \nabla F(\boldsymbol{x})\|_2^2] \leq \sigma_0^2 + \sigma_1^2 \|\nabla F(\boldsymbol{x})\|_2^p. \tag{7}$$

When $p = 2$, the $p$-affine variance (Assumption C.3) is reduced to the affine variance (Assumption C.2). When $\sigma_1 = 0$ or $p = 0$, the $p$-affine variance (Assumption C.3) is reduced to the bounded variance (Assumption C.1). The affine variance (Assumption C.2) was originally studied in (Bertsekas & Tsitsiklis, 2000) to analyze the convergence behavior of SGD. It was later applied to analyze AdaGrad (Faw et al., 2022; Wang et al., 2023a) and a simplified Adam (Wang et al., 2024).

To the best of our knowledge, Assumption B.3 and Assumption C.3 are the weakest assumptions for analyzing the convergence of Adam among the existing literatures.

In addition, we also list the following required conditions.

**Condition 1** *At any $t$-th iteration in Algorithm 1, the gradients satisfy $\sqrt{\frac{1}{T} \sum_{t=1}^{T} \|\nabla F(\boldsymbol{x}_t)\|_2^2} \leq \frac{C_0}{T} \sum_{t=1}^{T} \|\nabla F(\boldsymbol{x}_t)\|_2$ with $1 \leq C_0 \ll \sqrt{T}$.*

**Condition 2** *At any $t$-th iteration in Algorithm 1, the coordinates of the update in Adam , i.e., $u_t = \frac{|\boldsymbol{m}_t^{(j)}|}{\sqrt{\boldsymbol{v}_t^{(j)}} + \epsilon} (j \in [d])$, are independently and identically distributed (i.i.d), and the mean update is bounded away from zero, $\bar{u}_t = \frac{1}{d} \sum_{j=1}^{d} \frac{|\boldsymbol{m}_t^{(j)}|}{\sqrt{\boldsymbol{v}_t^{(j)}} + \epsilon} > 0$.*

**Condition 3** *At any $t$-th iteration in Algorithm 1, the gradient satisfies $\|\nabla F(\boldsymbol{x}_t)\|_1 = \frac{\sqrt{d}}{C_1} \|\nabla F(\boldsymbol{x}_t)\|_2$ with $1 \leq C_1 \ll \sqrt{d}$.*

Condition 1 is easily satisfied when the gradients $\|\nabla F(\boldsymbol{x}_t)\|_2$ decrease at a rate of $\Theta\left(\frac{1}{t^\alpha}\right)$ for all $0 < \alpha < \frac{1}{2}$, and the number of iterations is sufficiently large. We summarize this in the following proposition.

**Proposition 1** *If $\|\nabla F(\boldsymbol{x}_t)\|_2$ decreases at the rate of $\Theta\left(\frac{1}{t^\alpha}\right)$ for $0 \leq \alpha < \frac{1}{2}$ and $T \geq 8$, then it holds that*

$$\frac{\frac{1}{T} \sum_{t=1}^{T} \|\nabla F(\boldsymbol{x}_t)\|_2^2}{\left(\frac{1}{T} \sum_{t=1}^{T} \|\nabla F(\boldsymbol{x}_t)\|_2\right)^2} \leq \mathcal{O}\left(\frac{2(1-\alpha)^2}{1-2\alpha}\right). \tag{8}$$

Arjevani et al. (2023) has demonstrated that the optimal convergence rate of $\|\nabla F(\boldsymbol{x}_t)\|_2$ in non-convex stochastic optimization is $\mathcal{O}\left(\frac{1}{T^{1/4}}\right)$, which lies in the range $[0, \frac{1}{2})$. We choose a sufficiently large $T$ in practice, meaning $T$ is greatly larger than $\frac{2(1-\alpha)^2}{1-2\alpha}$. Therefore, Condition 1 commonly holds in practice.

Condition 2 commonly holds in practice, and we empirically validated it in our experiments, as shown in Figure 1. Specifically, we employed Adam to train ResNet-50 on ImageNet and GPT-2 (350M) on Open-WebText. During training, we recorded $\boldsymbol{m}_t^{(j)}/\sqrt{\boldsymbol{v}_t^{(j)}}$ for each coordinate in certain layers. To verify whether $|\boldsymbol{m}_t^{(j)}|/\sqrt{\boldsymbol{v}_t^{(j)}}$ for each coordinate is drawn from the same distribution, we used the two-sample Kolmogorov-Smirnov (KS) test. In this test, two groups of 10,000 samples were uniformly drawn from all coordinates of the layer, and these groups were used to run the two-sample KS test. We repeated this procedure 1,000 times and reported the mean $p$-value. As illustrated in Figure 1, the mean $p$-value is significantly larger than the significance level of 0.05, strongly suggesting that $\boldsymbol{m}_t^{(j)}/\sqrt{\boldsymbol{v}_t^{(j)}}$ for each coordinate in the layers is independently drawn from an identical distribution. It can also be observed that the mean value of $\hat{\boldsymbol{m}}_t^{(j)}/(\sqrt{\hat{\boldsymbol{v}}_t^{(j)}} + 10^{-8})$ stays almost stable and remains bounded away from zero during training.

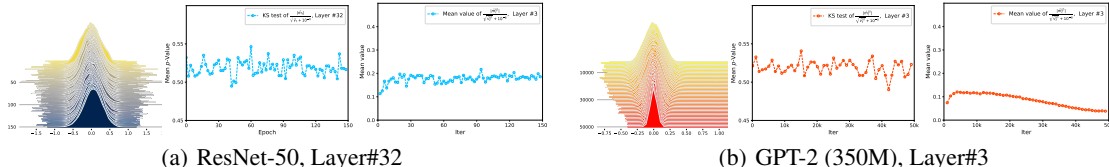

(a) ResNet-50, Layer#32        (b) GPT-2 (350M), Layer#3

Figure 1: The distribution, the two-sample Kolmogorov-Smirnov test and the mean value of $\hat{\boldsymbol{m}}_t^{(j)}/(\sqrt{\hat{\boldsymbol{v}}_t^{(j)}} + 10^{-8})$ across coordinates of (a) Layer#32.conv.weight in ResNet-50 during training with Adam on ImageNet for 150 epochs, and (b) Layer#3.self-attention.in-proj-weight in GPT-2 (350M) during training with Adam on OpenWebText for 5,000 iterations. In this test, two groups of 10,000 samples were uniformly drawn from all coordinates of the layer, and these groups were used to run the two-sample KS test. We repeated this procedure 1,000 times and reported the mean $p$-value. The $p$-values significantly exceed the 0.05 threshold, strongly indicating that the values of $\hat{\boldsymbol{m}}_t^{(j)}/(\sqrt{\hat{\boldsymbol{v}}_t^{(j)}} + 10^{-8})$ are independently drawn from the identical distribution. It can also be observed that the mean value of $\hat{\boldsymbol{m}}_t^{(j)}/(\sqrt{\hat{\boldsymbol{v}}_t^{(j)}} + 10^{-8})$ stays almost stable and remains bounded away from zero during training.

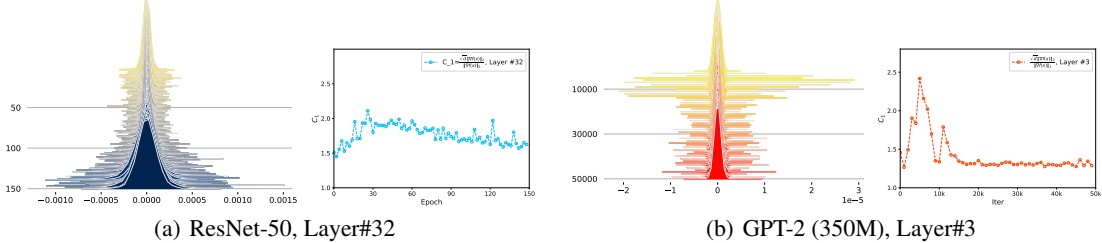

(a) ResNet-50, Layer#32        (b) GPT-2 (350M), Layer#3

Figure 2: The distribution and $C_1 = \frac{\sqrt{d}\|\nabla f(\boldsymbol{x}_t)\|_2}{\|\nabla f(\boldsymbol{x}_t)\|_1}$ for gradients across coordinates of (a) Layer#32.conv.weight in ResNet-50 during training with Adam on ImageNet for 150 epochs, and (b) Layer#3.self-attention.in-proj-weight in GPT-2 (350M) during training with Adam on OpenWebText for 50,000 iterations. Throughout training, $C_1$ remains consistently below 3, which is significantly smaller than $\sqrt{d}$, where $d$ represents the number of coordinates in the layers.

Condition 3 commonly holds in practice, and we also empirically validated it in our experiments, as shown in Figure 2. Specifically, we employed Adam to train ResNet-50 on ImageNet and GPT-2 (350M) on OpenWebText. During training, we recorded the gradient $\nabla f(\boldsymbol{x}_t)$ for each coordinate in selected layers. Subsequently, we computed $C_1 = \frac{\sqrt{d}\|\nabla f(\boldsymbol{x}_t)\|_2}{\|\nabla f(\boldsymbol{x}_t)\|_1}$. As shown in Figure 2, $C_1$ consistently remains below 3 throughout training, which is significantly smaller than $\sqrt{d}$, where $d$ represents the number of coordinates in the layers. This observation can be attributed to the fact that the coordinates of $\nabla f(\boldsymbol{x}_t)$ tend to be densely clustered during training, as also depicted in Figure 2.

## 3 MAIN RESULT

We first state the preliminary result in Theorem 2 under Assumptions A, B.3 and C.3. Then, we derive the more comprehensive convergence bound of Adam with Conditions 1, 2 and 3 in Corollary 3.

**Theorem 2** *Let $\{x_t\}_{t=1}^T$ be generated by Algorithm 1. Suppose that Assumptions A, B.3, and C.3, along with Condition 1, hold. Define $u_t^{(j)} := |\boldsymbol{m}_t^{(j)}|/(\sqrt{\boldsymbol{v}_t^{(j)}} + \epsilon)$, $R := (1 - \beta_1)/\sqrt{(1 - \beta_2)(1 - \frac{\beta_1^2}{\beta_2})}$, $\hat{L} := L_0 + (1 - q)L_1$, and $\hat{\sigma} := \sigma_0 + \sqrt{\frac{2-p}{2}}$. Choose $\beta_1 < \sqrt{\beta_2}$. Then, it holds for any $T \in \mathbb{N}^+$,*

$$\frac{1}{T}\left(\sum_{t=1}^T \mathbb{E}[\|\boldsymbol{u}_t \circ \nabla F(\boldsymbol{x}_t)\|_1] - \left(\frac{\gamma R^2 dqL_1}{2} + 2C_0R\sqrt{d}\sigma_1\sqrt{p(1-\beta_1)} + \frac{2\gamma R^2 dqL_1}{1-\beta_1}\right)\sum_{t=1}^T \mathbb{E}[\|\nabla F(\boldsymbol{x}_t)\|_2]\right) \tag{9}$$

$$\leq \frac{F(\boldsymbol{x}_1) - F^*}{\gamma T} + \frac{2R\sqrt{d}\|\nabla F(\boldsymbol{x}_1)\|_2}{T(1-\beta_1)} + 2\sqrt{1-\beta_1}R\sqrt{d}\hat{\sigma} + \frac{2\gamma R^2 d\hat{L}}{1-\beta_1} + \frac{\gamma R^2 d\hat{L}}{2}.$$

**Corollary 3** *Let $\{x_t\}_{t=1}^T$ be generated by Algorithm 1. Suppose Assumptions A, B.3 and C.3, along Conditions 1,2 and 3, hold. Define $u_t := |\hat{m}_t^{(j)}|/(\sqrt{\hat{v}_t^{(j)}} + \epsilon)$, $R := {}^{1-\beta_1}/\sqrt{(1-\beta_2)(1-\beta_1^2/\beta_2)}$, $\hat{L} := L_0 + (1-q)L_1$, and $\hat{\sigma} := \sigma_0 + \sqrt{\frac{2-p}{2}}$.*

*Case 1: General Setting with no Access to Oracles*

*Choose $\gamma = \frac{C_2}{T^{3/4}d^{1/2}}$, $\beta_1 < \sqrt{\beta_2}$, $1 - \beta_1 = \frac{C_3}{T^{1/2}}$, and $0 < \bar{v} \leq \min_t \mathbb{E}[u_t^{(j)}]$. Then, it holds for any $T \in \mathbb{N}^+$ and $T \geq (\frac{4C_1 C_2 R^2 q L_1}{C_3 \bar{v}} + \frac{4C_0 C_1 \sqrt{C_3} R \sigma_1 \sqrt{p}}{\bar{v}} + (\frac{C_1 C_2 R^2 q L_1}{\bar{v}})^{1/3})^4$,*

$$\frac{1}{T}\sum_{t=1}^T \mathbb{E}[\|\nabla F(\boldsymbol{x}_t)\|_2] \leq \frac{C_1}{\bar{v}}\left(\frac{2(F(\boldsymbol{x}_1) - F(\boldsymbol{x}^*))}{C_2 T^{1/4}} + \frac{4R\|\nabla F(\boldsymbol{x}_1)\|_2}{C_3 T^{1/2}} + \frac{4C_3 R\hat{\sigma}}{T^{1/4}} + \frac{4C_2 R^2 \hat{L}}{C_3 T^{1/4}} + \frac{C_2 R^2 \hat{L}}{T^{3/4}}\right). \quad (10)$$

*Case 2: Lowest-Bound Setting with Access to Oracles*

*Choose $\hat{C}_2 = \frac{(F(\boldsymbol{x}_1) - F^*)^{3/4}}{2^{1/4} R\hat{\sigma}^{1/2}\hat{L}^{1/4}}$, $\hat{C}_3 = \frac{2^{1/2}\hat{L}^{1/2}(F(\boldsymbol{x}_1) - F^*)^{1/2}}{\hat{\sigma}}$, $\gamma = \frac{\hat{C}_2}{T^{3/4}d^{1/2}}$, $\beta_1 < \sqrt{\beta_2}$, $1 - \beta_1 = \frac{\hat{C}_3}{T^{1/2}}$, and $0 < \bar{v} \leq \min_t \mathbb{E}[u_t^{(j)}]$. Then, for any $T \in \mathbb{N}^+$ and $T \geq (\frac{4C_1 \hat{C}_2 R^2 q L_1}{\hat{C}_3 \bar{v}} + \frac{4C_0 C_1 \sqrt{\hat{C}_3} R \sigma_1 \sqrt{p}}{\bar{v}} + (\frac{C_1 \hat{C}_2 R^2 q L_1}{\bar{v}})^{1/3})^4$, it reaches the lowest bound, i.e.,*

$$\frac{1}{T}\sum_{t=1}^T \mathbb{E}[\|\nabla F(\boldsymbol{x}_t)\|_2] \leq \frac{C_1}{\bar{v}}\left(\frac{512^{1/4} R\hat{\sigma}^{1/2}\hat{L}^{1/4}(F(\boldsymbol{x}_1) - F^*)^{1/4}}{T^{1/4}} + \frac{4R\|\nabla F(\boldsymbol{x}_1)\|_2}{\hat{C}_3 T^{1/2}} + \frac{\hat{C}_2 R^2 \hat{L}}{T^{3/4}}\right). \quad (11)$$

We have some findings from Theorem 3 below.

**Finding 1.** To the best of our knowledge, we are the first to formally analyze Adam under the weak assumptions of generalized non-uniform $(L_0, L_1, q)$-smoothness (Assumption B.3) and $p$-affine variance (Assumption C.3). We also prove that $\frac{1}{T}\sum_{t=1}^T \mathbb{E}[\|\nabla F(\boldsymbol{x}_t)\|_2]$ for Adam achieves a tighter bound of $\mathcal{O}\left(\frac{1}{T^{1/4}}\right)$, compared to the previous $\mathcal{O}\left(\frac{\ln T}{T^{1/4}}\right)$ (Chen et al., 2018; Défossez et al., 2020; Chen et al., 2022; Li et al., 2023). Furthermore, earlier works demonstrated that Adam's convergence rates were dependent on the model dimensionality $d$ and the numerical-stability $\epsilon$ (Chen et al., 2018; Défossez et al., 2020; Chen et al., 2022; Li et al., 2023), which makes them unsuitable to analyze large-scale LLM training. However, as shown in Corollary 3, we prove that Adam achieves dimension-free and $\epsilon$-free convergence, similar to SGD (Bottou et al., 2018). Notably, previous studies required a learning rate of $\gamma = 1 - \mathcal{O}(\frac{1}{T})$ to reach the optimal convergence rate of $\mathcal{O}\left(\frac{\ln T}{T^{1/4}}\right)$. This causes $\boldsymbol{v}_t$ to closely resemble a plain average of the past $T$ squared gradients, reducing Adam almost to AdaGrad (Défossez et al., 2020). By contrast, we only require the learning rate to satisfy $\gamma = 1 - \mathcal{O}(\frac{1}{T^{3/4}})$, which better aligns with Adam's original design.

**Finding 2.** The momentum coefficients $\beta_1$ and $\beta_2$ play a crucial role in Adam's convergence. As shown in Theorem 2, when $\beta_1 = \beta_2 = 0$, Adam reduces to signSGD, which converges at best to a bounded region where $\frac{1}{T}\sum_{t=1}^T \mathbb{E}[\|\nabla F(\boldsymbol{x}_t)\|_2] \leq \mathcal{O}(\frac{1}{T^{1/4}} + \sigma_0)$. This result aligns with prior work (Bernstein et al., 2018). In contrast, sufficiently large values of $\beta_1$ and $\beta_2$ ensure Adam achieves the convergence rate of $\mathcal{O}(\frac{1}{T^{1/4}})$. For SGD, however, momentum has minimal impact on the optimal theoretical convergence rate, as both momentum-SGD and vanilla SGD converge at $\mathcal{O}(\frac{1}{T^{1/4}})$ (Bottou et al., 2018; Liu et al., 2020).

**Finding 3.** Case 2 in Corollary 3 states that Adam's learning rate must satisfy $\gamma = \mathcal{O}(\frac{1}{\sqrt{d}})$ to achieve the optimal convergence rate. This implies that, with fixed other hyperparameters, larger model sizes require smaller optimal learning rates. This observation has been empirically validated by practitioners training the

## 4 PROOF SKETCH

In this section, we present the core ideas underlying the convergence proofs for Theorem 2 and Case 2 of Corollary 3. The proof ideas for Case 1 of Corollary 3 is similar, and thus, we omit the details for simplicity.

Our main contribution lies in opening up a new approach to proving the convergence of Adam. All existing theoretical convergence proofs follow a path-dependent approach, treating Adam as a preconditioned SGD with momentum, as originally presented in the Adam paper (Kingma & Ba, 2015). Specifically, the update rule is defined as: *i.e.*, $\boldsymbol{x}_t + 1 = \boldsymbol{x}_t - \frac{\gamma_t}{\sqrt{\boldsymbol{v}_t} + \epsilon} \circ \boldsymbol{m}_t$, where $\sqrt{\boldsymbol{v}_t} + \epsilon$ is used to precondition $\boldsymbol{m}_t$, and the effective learning rate is $\frac{\gamma_t}{\sqrt{\boldsymbol{v}_t} + \epsilon}$. This approach, however, encounters two intractable issues: $(i)$ the effective learning rate $\frac{\gamma_t}{\sqrt{\boldsymbol{v}_t} + \epsilon}$ is not necessarily monotone-decreasing, and $(ii)$ the random variable $\boldsymbol{v}_t$ is not independent of $\boldsymbol{g}_t$ or $\boldsymbol{m}_t$. To address these challenges, the proofs in previous works became complicated, lengthy, and opaque, making them difficult to verify and extend. In contrast, we treat Adam as a whole stochastic sign-like descent algorithm, *i.e.*, $\boldsymbol{x}_{t+1} = \boldsymbol{x}_t - \gamma_t \frac{|\boldsymbol{m}_t|}{\sqrt{\boldsymbol{v}_t} + \epsilon} \circ \mathrm{Sign}(\boldsymbol{m}_t)$ where we consider the term $\frac{|\boldsymbol{m}_t|}{\sqrt{\boldsymbol{v}_t} + \epsilon}$ as a single random variable. This transformation not only circumvents the problems mentioned above but also simplifies the proof process. We now provide a sketch of the proof.

Under Assumption B.3, we obtain (details please refer to Lemma 1 in the Appendix):

$$\mathbb{E}[F(\boldsymbol{x}_{t+1})] \leq F(\boldsymbol{x}_t) + \mathbb{E}[\langle \nabla F(\boldsymbol{x}_t), \boldsymbol{x}_{t+1} - \boldsymbol{x}_t \rangle] + \frac{L_0 + L_1 \|\nabla F(\boldsymbol{x}_t)\|_2^q}{2} \mathbb{E}[\|\boldsymbol{x}_{t+1} - \boldsymbol{x}_t\|_2^2]. \tag{12}$$

Defining $\boldsymbol{u}_t := \frac{|\boldsymbol{m}_t|}{\sqrt{\boldsymbol{v}_t} + \epsilon}$, the update rule becomes $\boldsymbol{x}_{t+1} = \boldsymbol{x}_t - \gamma \frac{\boldsymbol{m}_t}{\sqrt{\boldsymbol{v}_t} + \epsilon} = \boldsymbol{x}_t - \gamma \frac{|\boldsymbol{m}_t|}{\sqrt{\boldsymbol{v}_t} + \epsilon} \circ \frac{\boldsymbol{m}_t}{|\boldsymbol{m}_t|} = \boldsymbol{x}_t - \gamma \boldsymbol{u}_t \circ \mathrm{Sign}(\boldsymbol{m}_t)$. We further have

$$\mathbb{E}[F(\boldsymbol{x}_{t+1})] \leq F(\boldsymbol{x}_t) - \gamma \mathbb{E}[\|\boldsymbol{u}_t \nabla F(\boldsymbol{x}_t)\|_1] + \underbrace{\gamma \mathbb{E}[\langle \nabla F(\boldsymbol{x}_t), \boldsymbol{u}_t \circ (\mathrm{Sign}(\nabla F(\boldsymbol{x}_t)) - \mathrm{Sign}(\boldsymbol{m}_t)) \rangle]}_{\mathcal{T}_1}$$

$$+ \underbrace{\frac{\gamma^2 (L_0 + L_q \|\nabla F(\boldsymbol{x}_t)\|_2^q)}{2} \mathbb{E}[\|\boldsymbol{u}_t\|_2^2]}_{\mathcal{T}_2}. \tag{13}$$

Next, we define $R := \frac{1 - \beta_1}{\sqrt{(1 - \beta_2)(1 - \beta_1^2/\beta_2)}}$, and by applying Lemma 2 in the appendix, we obtain that $\boldsymbol{u}_t^{(j)} \leq R$. Furthermore, Lemma 3 in the appendix indicates $\mathbb{E}[|\mathrm{Sign}(\nabla F(\boldsymbol{x}_t^{(j)})) - \mathrm{Sign}(\boldsymbol{m}_t^{(j)})|] \leq 2 \frac{\mathbb{E}[|\nabla F(\boldsymbol{x}_t^{(j)}) - \boldsymbol{m}_t^{(j)}|]}{|\nabla F(\boldsymbol{x}_t^{(j)})|}$, which leads to $\mathcal{T}_1 \leq 2\gamma R \sqrt{d} \mathbb{E}[\|\nabla F(\boldsymbol{x}_t) - \boldsymbol{m}_t\|_2]$.

Employing the bound $\boldsymbol{u}_t^{(j)} \leq R$ above and applying Young's inequality, we obtain $\mathcal{T}_2 \leq \frac{\gamma^2 R^2 d(L_0 + L_1((1-q) + q\mathbb{E}[\|\nabla F(\boldsymbol{x}_t)\|_2]))}{2}$.

By taking the expectation over the first to the $(T-1)$-th iteration, and then summing and rearranging the terms, we obtain:

$$\frac{1}{T} \sum_{t=1}^{T} \mathbb{E}[\|\boldsymbol{u}_t \circ \nabla F(\boldsymbol{x}_t)\|_1] - \frac{\gamma R^2 q L_1 d}{2T} \sum_{t=1}^{T} \mathbb{E}[\|\nabla F(\boldsymbol{x}_t)\|_2]$$

$$\leq \frac{F(\boldsymbol{x}_1) - F^*}{\gamma T} + \frac{2R\sqrt{d}}{T} \sum_{t=1}^{T} \mathbb{E}[\|\boldsymbol{m}_t - \nabla F(\boldsymbol{x}_t)\|_2] + \frac{\gamma R^2 d(L_0 + (1-q)L_1)}{2T}. \tag{14}$$

We now divide-and-conquer prove that

$$\frac{1}{T}\sum_{t=1}^{T}\mathbb{E}\left[\|\boldsymbol{m}_t - \nabla F(\boldsymbol{x}_t)\|_2\right] \leq \frac{\|\nabla F(\boldsymbol{x}_1)\|_2}{T(1-\beta_1)} + \sqrt{1-\beta_1}\left(\sigma_0 + \sqrt{\frac{2-p}{2}}\sigma_1\right)$$

$$+ C_0\sigma_1\sqrt{p(1-\beta_1)} \cdot \frac{1}{T}\sum_{t=1}^{T}\|\nabla F(x_t)\|_2 \tag{15}$$

$$+ \frac{\gamma R\sqrt{d}(L_0 + (1-q)L_1)}{1-\beta_1} + \frac{\gamma R\sqrt{d}qL_1}{1-\beta_1} \cdot \frac{1}{T}\sum_{t=1}^{T}\mathbb{E}[\|\nabla F(\boldsymbol{x}_t)\|_2]$$

Then, we have

$$\frac{1}{T}\left(\sum_{t=1}^{T}\mathbb{E}[\|\boldsymbol{u}_t \circ \nabla F(\boldsymbol{x}_t)\|_1] - \left(\frac{\gamma R^2 dqL_1}{2} + 2C_0R\sqrt{d}\sigma_1\sqrt{p(1-\beta_1)} + \frac{2\gamma R^2 dqL_1}{1-\beta_1}\right)\sum_{t=1}^{T}\mathbb{E}[\|\nabla F(\boldsymbol{x}_t)\|_2]\right)$$

$$\leq \frac{F(\boldsymbol{x}_1) - F^*}{\gamma T} + \frac{2R\sqrt{d}\|\nabla F(\boldsymbol{x}_1)\|_2}{T(1-\beta_1)} + 2\sqrt{1-\beta_1}R\sqrt{d}\hat{\sigma} + \frac{2\gamma R^2 d\hat{L}}{1-\beta_1} + \frac{\gamma R^2 d\hat{L}}{2T}. \tag{16}$$

where $\hat{L} := L_0 + (1-q)L_1$, and $\hat{\sigma} := \sigma_0 + \sqrt{\frac{2-p}{2}}$.

By choosing $\bar{v} = \min_t \mathbb{E}[\boldsymbol{u}_t^{(j)}]$ and applying Condition 2 and Condition 3, we obtain

$$\sum_{t=1}^{T}\mathbb{E}[\|\boldsymbol{u}_t \circ \nabla F(\boldsymbol{x}_t)\|_1] = \sum_{t=1}^{T}\mathbb{E}[\boldsymbol{u}_t^{(j)}]\mathbb{E}[\|\nabla F(\boldsymbol{x}_t)\|_1] \geq \bar{v}\sum_{t=1}^{T}\mathbb{E}[\|(\nabla F(\boldsymbol{x}_t))\|_1] = \frac{\bar{v}\sqrt{d}}{C_1}\sum_{t=1}^{T}\mathbb{E}[\|(\nabla F(\boldsymbol{x}_t))\|_2]. \tag{17}$$

Using generalized Young's inequality, we minimize the bottleneck terms to achieve the lowest bound on the right-hand side of Eq. (16), *i.e.*,

$$\frac{C_1(F(\boldsymbol{x}_1) - F^*)}{\gamma T\sqrt{d}} + 2C_1\sqrt{1-\beta_1}R\hat{\sigma} + \frac{2C_1\gamma R^2\sqrt{d}\hat{L}}{1-\beta_1} \geq \frac{512^{1/4}C_1R\hat{\sigma}^{1/2}\hat{L}^{1/4}(F(\boldsymbol{x}_1) - F^*)}{T^{1/4}}, \tag{18}$$

where the lowest bound achieved if and only if $\gamma = \frac{(F(\boldsymbol{x}_1)-F^*)^{3/4}}{2^{1/4}T^{3/4}d^{1/2}R\hat{\sigma}^{1/2}\hat{L}^{1/4}}$ and $1-\beta_1 = \frac{2^{1/2}\hat{L}^{1/2}(F(\boldsymbol{x}_1)-F^*)^{1/2}}{T^{1/2}\hat{\sigma}}$.

Now, choosing $\hat{C}_2 = \frac{(F(\boldsymbol{x}_1)-F^*)^{3/4}}{2^{1/4}R\hat{\sigma}^{1/2}\hat{L}^{1/4}}$, $\hat{C}_3 = \frac{2^{1/2}\hat{L}^{1/2}(F(\boldsymbol{x}_1)-F^*)^{1/2}}{\hat{\sigma}}$, $\gamma = \frac{\hat{C}_2}{T^{3/4}d^{1/2}}$, $1-\beta_1 = \frac{\hat{C}_3}{T^{1/2}}$ and setting $T \geq \left(\frac{4C_0C_1\hat{C}_3^{1/2}p^{1/2}R\sigma_1}{\bar{v}} + \frac{4C_1\hat{C}_2R^2qL_1}{\hat{C}_3\bar{v}} + \left(\frac{C_1\hat{C}_2R^2qL_1}{\bar{v}}\right)^{1/3}\right)^4$, we arrive Case 2 of Corollary 3, *i.e.*,

$$\frac{1}{T}\sum_{t=1}^{T}\mathbb{E}[\|\nabla F(\boldsymbol{x}_t)\|_2] \leq \frac{C_1}{\bar{v}}\left(\frac{512^{1/4}R\hat{\sigma}^{1/2}\hat{L}^{1/4}(F(\boldsymbol{x}_1)-F^*)^{1/4}}{T^{1/4}} + \frac{2C_1R\|\nabla F(\boldsymbol{x}_1)\|_2}{\hat{C}_3T^{1/2}} + \frac{\hat{C}_2R^2\hat{L}}{T^{3/4}}\right). \tag{19}$$

## 5 CONCLUSION

This work breaks with convention and provides a pioneering reinterpretation of Adam as a sign-like descent algorithm to analyze the convergence, simplifying its theoretical analysis and addressing limitations of the traditional preconditioned perspective. By treating $\frac{|\boldsymbol{m}_t|}{\sqrt{\boldsymbol{v}_t}+\epsilon}$ as a unified random variable, this is the first time that it have been proved that Adam dimension-freely and $\epsilon$-freely achieves the optimal convergence rate of $\mathcal{O}(\frac{1}{T^{1/4}})$ under the weak assumptions of generalized $(L_0, L_1, q)$-smoothness and affine variance.

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

# Appendix

**The Use of Large Language Models (LLMs).** The use of LLMs in the preparation of this work is confined to that of a polishing and assistive tool. They were not involved in the core intellectual processes of research ideation, hypothesis formation, or theoretical derivation. Their role was limited to tasks such as checking grammar, refining sentence structure, and improving the clarity of text that was entirely conceived and drafted by the human authors.

## A    RELATE WORK

There is a large amount of works on the theoretical analysis of stochastic descents algorithms. In this section, we list the most related references and make comparison with our work.

**Convergence with Weak Assumptions.** Bertsekas & Tsitsiklis (2000) first theoretically analyze SGD under the assumption of affine variances, obtaining an asymptotic convergence result. Until 2018, Bottou et al. (2018) proved the non-asymptotic convergence rate of $\|\nabla F(x)\|_2$ for SGD up to $\mathcal{O}\left(\frac{\text{poly}(\ln T)}{T^{1/4}}\right)$, which matched its provable rate with the bounded variance condition. In terms of adaptive optimizers, Faw et al. (2022) investigated the convergence rate of AdaGrad-Norm with the affine variance, and proved the rate could achieve $\mathcal{O}\left(\frac{\text{poly}(\ln T)}{T^{1/4}}\right)$ as well when $\sigma_1 = \mathcal{O}\left(\frac{1}{\sqrt{T}}\right)$. Wang et al. (2023a) proved the AdaGrad-Norm obtained a similar convergence rate with no restriction over $\sigma_1$, and it further demonstrate vanilla AdaGrad could also achieve the same convergence rate under a stronger assumption of coordinate-wise affine variances. Meanwhile, Attia & Koren (2023) provided a probabilistic convergence rate for AdaGrad-Norm. Noted that Shi et al. (2021) and Zhang et al. (2022) respectively proved random-shuffled AMSProp and Adam will converge to the neighbourhood of stationary points with the rate $\mathcal{O}\left(\frac{\text{poly}(\ln E)}{E^{1/4}} + \sigma_0\right)$ where $E$ is the number of epochs rather than iterations under the affine growth condition that is equivalent to the affine variance condition, which is much slower than the optimal rate. Recently, Hong & Lin (2023) provably demonstrate the convergence rate of vanilla Adam in high probability perspective, but it only works with the stronger coordinate-wise affine variance.

Zhang et al. (2019) first introduced the unciform $(L_0, L_1)$-smooth condition to theoretically explain why Clipped-SGD converges faster than vanilla SGD, and they also empirically verified that local smoothness indeed varies with the norm of gradients during DNN training. Zhang et al. (2020) posits that it is also equivalent to an affine form of the gradient norm for the first-order differentiable function. Then, this relaxed assumption is extended to analyzing clipping-SGD with momentum Zhang et al. (2020), distributionally robust optimization Jin et al. (2021), normalized SGD with momentum Hübler et al. (2024), generalized signSGD Crawshaw et al. (2022). Recently, Wang et al. (2023c) theoretically analyzing random-shuffled Adam under this condition, but its convergence rate is provable $\mathcal{O}\left(\frac{\text{poly}(\ln E)}{E^{1/4}}\right)$ where $E$ is the number of epoch, just like Zhang et al. (2022). Li et al. (2023) further extended the linear $(L_0, L_1)$-smooth to the generalized polynomial version, and proved that Adam will converged to $\mathcal{O}\left(\frac{\text{poly}(\ln T)}{T^{1/4}}\right)$ with the weaker assumption. However, the bound heavily relies on a large $\epsilon$, but Adam with a large $\epsilon$ is essentially similar to SGD and loses the nature of adaptivity.

Furthermore, Faw et al. (2023); Wang et al. (2023b) theoretically analyze AdaGrad under the weak assumptions of both affine variance and non-uniform $(L_0, L_1)$-smoothness, also obtaining the rate of $\mathcal{O}\left(\frac{\text{poly}(\ln T)}{T^{1/4}}\right)$. Wang et al. (2024) also provable provide a high probability convergence of a simplified Adam-Norm with the both weak assumptions. More recently, Hong & Lin (2025) prove the convergence rate of vanilla Adam with both the affine variance condition and the $(L_0, L_1)$-smooth condition, but its proof have a fatal error that may vacuum the validity (The constant and $T$-independent $G$ is the premise of the theoretical analysis,

but $1 - \beta_2 = \mathcal{O}(\frac{1}{T})$, $1 - \beta_2 = \frac{c}{T}$, Eq. (41), Eq. (56) and Eq. (58) in the proof suggest that $\|\bar{\boldsymbol{g}}_{t+1}\|^2$ should be larger than $\mathcal{O}(T)$. ).

In comparison to the convergence analyses above, we are the first to formally analyze vanilla under the weak assumptions of the generalized affine variance and the generalized unciform $(L_0, L_1)$-smooth, achieving a tighter bound of $\mathcal{O}\left(\frac{1}{T^{1/4}}\right)$ rather than $\mathcal{O}\left(\frac{\text{poly}(\ln T)}{T^{1/4}}\right)$.

**Convergence of Sign Descent.** Sign-based algorithms that simply exploits the signs of gradients could date back to RPROP (Riedmiller & Braun (1993)). Seide et al. (2014); Ström (2015) proposed 1-bit SGD and empirically demonstrate it achieve good performance while dramatically reducing the communication costs in distributed system. The non-stochastic convergence proof of signSGD was first analyzed in (Karimi et al. (2016)) under the Polyak-Łojasiewicz condition. Then, Bernstein et al. (2018) systematically establish the convergence rate of signSGD in stochastic and non-convex scenario, but it required an increasing batch size up to $O(\sqrt{n})$ where $n$ is the number of samples to guarantee convergence. Then, Sun et al. (2023) first proved that momentum can ensures the convergence of signSGD without increasing batch size. Chen et al. (2023b) employs an AutoML method to discover an effective optimizer, Lion, resembling signSGD with momentum, and demonstrate superior performance to Adam across diverse DNN models. Chen et al. (2023a) theoretically analyzed the efficacy of Lion but did not provide the convergence proof. Meanwhile, the original version of Adam, RMSProp Hinton et al. (2012), were developed from the sign-based Rprop. Balles & Hennig (2018) also found that sign descent algorithms has a deep connection with Adam. Recently, Kunstner et al. (2023; 2024) empirically showcase that the sign-like property of Adam is just the primary reason behind its superior performance for training DNNs. In short, signSGD has a close connection with Adam, but the existing convergence proofs of Adam were not built upon this connection. More recently, Muon Jordan et al. (2024), an extended matrix-sign optimizer, has demonstrated significant potential for training DNNs Liu et al. (2025); Shah et al. (2025), although its current implementation is confined to parameters with a 2D structure.

To the best of our knowledge, our work is the first to prove the convergence rate of Adam from the perspective of sign descent, and the proof, thereby, becomes considerably simple, compared to the previous theoretical proofs of Adam.

# B  THEORETICAL ANALYSIS

## B.1  PROOF OF PROPOSITION 2.1

**Proof.** It is known that

$$
\begin{aligned}
\frac{1}{T} \sum_{t=1}^{T} \|\nabla F(\boldsymbol{x}_t)\|_2^2 &= \Theta\left(\frac{1}{T} \sum_{t=1}^{T} \frac{1}{t^{2\alpha}}\right) \\
&\leq \mathcal{O}\left(\frac{1}{T} \int_{t=1}^{T} \frac{1}{t^{2\alpha}} dt\right) \\
&= \mathcal{O}\left(\frac{T^{1-2\alpha} - 1}{T(1 - 2\alpha)}\right) \\
&\leq \mathcal{O}\left(\frac{1}{1 - 2\alpha} \cdot \frac{1}{T^{2\alpha}}\right).
\end{aligned}
\tag{20}
$$

and

$$\left(\frac{1}{T}\sum_{t=1}^{T}\|\nabla F(\boldsymbol{x}_t)\|_2\right)^2 = \Theta\left(\left(\frac{1}{T}\sum_{t=1}^{T}\frac{1}{t^\alpha}\right)^2\right)$$

$$\geq \Omega\left(\left(\frac{1}{T}\int_{t=2}^{T+1}\frac{1}{t^\alpha}dt\right)^2\right) \tag{21}$$

$$\geq \Omega\left(\frac{1}{(1-\alpha)^2}\cdot\left(\frac{1}{T^{2\alpha}} - \frac{2^{1-\alpha}}{T^{1+\alpha}}\right)\right).$$

It is easy to verify that when $T \geq 2^{\frac{2-\alpha}{1-\alpha}}$, $\frac{2^{1-\alpha}}{T^{1+\alpha}} \leq \frac{1}{2}\cdot\frac{1}{T^{2\alpha}}$. Moreover, we can obtain $2^{\frac{2-\alpha}{1-\alpha}} \leq 8$ due to $0 \leq \alpha < \frac{1}{2}$. Hence, when $T \geq 8$, we have

$$\frac{\frac{1}{T}\sum_{t=1}^{T}\|\nabla F(\boldsymbol{x}_t)\|_2^2}{\left(\frac{1}{T}\sum_{t=1}^{T}\|\nabla F(\boldsymbol{x}_t)\|_2\right)^2} \leq \mathcal{O}\left(\frac{2(1-\alpha)^2}{1-2\alpha}\right). \tag{22}$$

## B.2 USEFUL LEMMAS

**Lemma 1** *Under Assumption B.3, for any $\boldsymbol{x}, \boldsymbol{y} \in \mathbb{R}^d$, the function obeys*

$$F(\boldsymbol{y}) \leq F(\boldsymbol{x}) + \langle\nabla F(\boldsymbol{x}), \boldsymbol{y} - \boldsymbol{x}\rangle + \frac{L_0 + L_1\|\nabla F(\boldsymbol{x})\|_2^q}{2}\|\boldsymbol{y} - \boldsymbol{x}\|_2^2. \tag{23}$$

**Proof.** For any $\boldsymbol{x}, \boldsymbol{y} \in \mathbb{R}^d$, we have

$$F(\boldsymbol{y}) = F(\boldsymbol{x}) + \int_0^1 \langle\nabla F(\boldsymbol{x} + t(\boldsymbol{y} - \boldsymbol{x})), \boldsymbol{y} - \boldsymbol{x}\rangle dt$$

$$= F(\boldsymbol{x}) + \langle\nabla F(\boldsymbol{x}), \boldsymbol{y} - \boldsymbol{x}\rangle + \int_0^1 \langle\nabla F(\boldsymbol{x} + t(\boldsymbol{y} - \boldsymbol{x})) - \nabla F(\boldsymbol{x}), \boldsymbol{y} - \boldsymbol{x}\rangle dt$$

$$\overset{(i)}{\leq} F(\boldsymbol{x}) + \langle\nabla F(\boldsymbol{x}), \boldsymbol{y} - \boldsymbol{x}\rangle + \int_0^1 \|\nabla F(\boldsymbol{x} + t(\boldsymbol{y} - \boldsymbol{x})) - \nabla F(\boldsymbol{x})\|_2\|\boldsymbol{y} - \boldsymbol{x}\|_2 dt \tag{24}$$

$$\overset{(ii)}{\leq} F(\boldsymbol{x}) + \langle\nabla F(\boldsymbol{x}), \boldsymbol{y} - \boldsymbol{x}\rangle + (L_0 + L_1\|\nabla F(\boldsymbol{x})\|_2^q)\|\boldsymbol{y} - \boldsymbol{x}\|_2^2\int_0^1 t d_t$$

$$= F(\boldsymbol{x}) + \langle\nabla F(\boldsymbol{x}), \boldsymbol{y} - \boldsymbol{x}\rangle + \frac{L_0 + L_1\|\nabla F(\boldsymbol{x})\|_2^q}{2}\|\boldsymbol{y} - \boldsymbol{x}\|_2^2,$$

where $(i)$ holds due to Cauchy-Schwarz inequality, and $(ii)$ holds due to Assumption 2.

**Lemma 2** *Let the sequences $\{\hat{\boldsymbol{m}}_t\}$ and $\{\hat{\boldsymbol{v}}_t\}$ be generated by Adam in Algorithm 1. If the moving average coefficients $\beta_1, \beta_2$ are constant and satisfy $\beta_1^2 < \beta_2$, then*

*(1) For any $j \in [d]$, it holds that*

$$\frac{|\hat{\boldsymbol{m}}_t^{(j)}|}{\sqrt{\hat{\boldsymbol{v}}_t^{(j)}} + \epsilon} \leq \frac{1 - \beta_1}{\sqrt{1 - \beta_2}\sqrt{1 - \frac{\beta_1^2}{\beta_2}}}. \tag{25}$$

*(2) The maximal value of $\frac{1-\beta_1}{\sqrt{1-\beta_2}\sqrt{1-\frac{\beta_1^2}{\beta_2}}}$ is lower bounded by* 1*, and upper bounded by* $\sqrt{\frac{1-\beta_1}{1-\beta_2}}$ *when* $\beta_1 \leq \beta_2$.

**Proof.** (1) Recalling Adam, we know

$$
\hat{\boldsymbol{m}}_t^{(j)} = \frac{1-\beta_1}{1-\beta_1^t} \sum_{k=1}^t \beta_1^{t-k} \boldsymbol{g}_k^{(j)}
$$

$$
\hat{\boldsymbol{v}}_t^{(j)} = \frac{1-\beta_2}{1-\beta_2^t} \sum_{k=1}^t \beta_2^{t-k} (\boldsymbol{g}_k^{(j)})^2. \tag{26}
$$

Then,

$$
\begin{aligned}
\frac{|\boldsymbol{m}_t^{(j)}|}{\sqrt{\boldsymbol{v}_t^{(j)}} + \epsilon} &\leq \frac{(1-\beta_1)\sqrt{1-\beta_2^t}}{(1-\beta_1^t)\sqrt{1-\beta_2}} \cdot \frac{|\sum_{k=1}^t \beta_1^{t-k} \boldsymbol{g}_k^{(j)}|}{\sqrt{\sum_{k=1}^t \beta_2^{t-k}(\boldsymbol{g}_k^{(j)})^2}} \\[2mm]
&\overset{(i)}{\leq} \frac{(1-\beta_1)\sqrt{1-\beta_2^t}}{(1-\beta_1^t)\sqrt{1-\beta_2}} \cdot \frac{\sum_{k=1}^t \beta_1^{t-k}|\boldsymbol{g}_k^{(j)}|}{\sqrt{\sum_{k=1}^t \beta_2^{t-k}(\boldsymbol{g}_k^{(j)})^2}} \\[2mm]
&\overset{(ii)}{\leq} \frac{(1-\beta_1)\sqrt{1-\beta_2^t}}{(1-\beta_1^t)\sqrt{1-\beta_2}} \cdot \frac{\sqrt{\sum_{k=1}^t \beta_2^{t-k}(\boldsymbol{g}_k^{(j)})^2}\sqrt{\sum_{k=1}^t \frac{\beta_1^{2(t-k)}}{\beta_2^{t-k}}}}{\sqrt{\sum_{k=1}^t \beta_2^{t-k}(\boldsymbol{g}_k^{(j)})^2}} \\[2mm]
&= \frac{(1-\beta_1)\sqrt{1-\beta_2^t}}{(1-\beta_1^t)\sqrt{1-\beta_2}} \cdot \sqrt{\sum_{k=1}^t \left(\frac{\beta_1^2}{\beta_2}\right)^{t-k}} \\[2mm]
&\overset{(iii)}{=} \frac{(1-\beta_1)\sqrt{1-\beta_2^t}}{(1-\beta_1^t)\sqrt{1-\beta_2}} \cdot \frac{\sqrt{1-\left(\frac{\beta_1^2}{\beta_2}\right)^t}}{\sqrt{1-\frac{\beta_1^2}{\beta_2}}} \\[2mm]
&\overset{(iv)}{\leq} \frac{1-\beta_1}{\sqrt{1-\beta_2}\sqrt{1-\frac{\beta_1^2}{\beta_2}}},
\end{aligned} \tag{27}
$$

where $(i)$ holds due to the fact $|\boldsymbol{a}^{(j)} + \boldsymbol{b}^{(j)}| \leq |\boldsymbol{a}^{(j)}| + |\boldsymbol{b}^{(j)}|$; $(ii)$ holds resulting from Cauchy-Schwarz inequality; $(iii)$ holds since $\beta_1^2 \leq \beta_2$; $(iv)$ holds owing to the fact that $1 - \frac{a^2}{b} \leq \frac{(1-a)^2}{1-b}$.

(2) The maximal value of $\frac{|\boldsymbol{m}_t^{(j)}|}{\sqrt{\boldsymbol{v}_t^{(j)}}}$ is lower bounded by

$$
\frac{1-\beta_1}{\sqrt{1-\beta_2}\sqrt{1-\frac{\beta_1^2}{\beta_2}}} \geq \frac{1-\beta_1}{1-\frac{1}{2}(\beta_2 + \frac{\beta_1^2}{\beta_2})} \geq \frac{1-\beta_1}{1-\beta_1} = 1, \tag{28}
$$

where the first and second inequality reaches the lower bound if and only $\beta_1 = \beta_2$.

When $\beta_1 \leq \beta_2$, the maximal value of $\frac{|\boldsymbol{m}_t^{(j)}|}{\sqrt{\boldsymbol{v}_t^{(j)}}}$ is upper bounded by

$$
\frac{1-\beta_1}{\sqrt{1-\beta_2}\sqrt{1-\frac{\beta_1^2}{\beta_2}}} \leq \frac{1-\beta_1}{\sqrt{1-\beta_2}\sqrt{1-\frac{\beta_1\beta_2}{\beta_2}}} = \frac{\sqrt{1-\beta_1}}{\sqrt{1-\beta_2}}. \tag{29}
$$

**Lemma 3** *For any random variable $\mathcal{Z}$ and a constant $C$, there exists*

$$\mathbb{E}[|\text{Sign}(\mathcal{Z}) - \text{Sign}(C)|] \leq \frac{2\mathbb{E}[|\mathcal{Z} - C|]}{|C|}. \tag{30}$$

**Proof.** Using Markov'equality, we direct obtain

$$\begin{aligned}
\mathbb{E}[|\text{Sign}(\mathcal{Z}) - \text{Sign}(C)|] =& 2\mathbb{P}[\mathbb{I}(\text{Sign}(\mathcal{Z}) \neq \text{Sign}(C))] \\
\leq& 2\mathbb{P}[|\mathcal{Z} - C| \geq |C|] \\
\leq& \frac{2\mathbb{E}[|\mathcal{Z} - C|]}{|C|}.
\end{aligned} \tag{31}$$

**Lemma 4** *Let $a, b > 0$ and $0 < \alpha < \beta$. If $x \geq (a + b^{\alpha/\beta})^{1/\alpha}$, then $\frac{a}{x^\alpha} + \frac{b}{x^\beta} \leq 1$. The bound is tight up to the factor of 2 since $\frac{(a+b^{\alpha/\beta})^{1/\alpha}}{2} \leq \max(a, b^{\alpha/\beta}) \leq (a + b^{\alpha/\beta})^{1/\alpha}$.*

**Proof.** Let $s = x^\alpha$ and $\gamma = \frac{\beta}{\alpha} \geq 1$, then the inequality becomes

$$\frac{a}{s} + \frac{b}{s^\gamma} \leq 1 \tag{32}$$

If $s^*$ is a solution of Eq. (32), it should satisfy

$$s^* \geq \max(a, b^{1/\gamma}). \tag{33}$$

When we set $s_+ = a + b^{1/\gamma}$, it is easy to verify that

$$\frac{a}{s_+} + \frac{b}{s_+^\gamma} = \frac{a}{a + b^{1/\gamma}} + \frac{b}{(a + b^{1/\gamma})^\gamma} \leq \frac{a}{a + b^{1/\gamma}} + \frac{b^{1/\gamma}}{a + b^{1/\gamma}} = 1. \tag{34}$$

On the other hand, it is also easy to verify that $s_- = \frac{a+b^{1/\gamma}}{2}$ does not satisfy Eq. (33), which means that $s_+$ is at most a factor of 2 worse than the smallest solution of Eq. (32), so $x_+ = (a + b^{\alpha/\beta})^{1/\alpha}$ is at most a factor of 2 worse than the smallest solution of $\frac{a}{x^\alpha} + \frac{b}{x^\beta} \leq 1$.

### B.3 PROOF OF THEOREM 2

**Proof.** Following Lemma 1 with $\boldsymbol{x}_{t+1} \to \boldsymbol{y}$ and $\boldsymbol{x}_t \to \boldsymbol{x}$, we have

$$F(\boldsymbol{x}_{t+1}) \leq F(\boldsymbol{x}_t) + \langle \nabla F(\boldsymbol{x}_t), \boldsymbol{x}_{t+1} - \boldsymbol{x}_t \rangle + \frac{L_0 + L_1\|\nabla F(\boldsymbol{x}_t)\|_2^q}{2}\|\boldsymbol{x}_{t+1} - \boldsymbol{x}_t\|_2^2. \tag{35}$$

Recalling the update rule $\boldsymbol{x}_{t+1} = \boldsymbol{x}_t - \gamma \frac{\hat{\boldsymbol{m}}_t}{\sqrt{\hat{\boldsymbol{v}}_t} + \epsilon} = \boldsymbol{x}_t - \gamma \frac{|\hat{\boldsymbol{m}}_t|}{\sqrt{\hat{\boldsymbol{v}}_t} + \epsilon} \circ \frac{\hat{\boldsymbol{m}}_t}{|\hat{\boldsymbol{m}}_t|} = \boldsymbol{x}_t - \gamma \boldsymbol{u}_t \circ \text{Sign}(\hat{\boldsymbol{m}}_t) = \boldsymbol{x}_t - \gamma \boldsymbol{u}_t \circ \text{Sign}(\boldsymbol{m}_t)$, we further obtain

$$\begin{aligned}
F(\boldsymbol{x}_{t+1}) \leq& F(\boldsymbol{x}_t) - \langle \nabla F(\boldsymbol{x}_t), \gamma \boldsymbol{u}_t \circ \text{Sign}(\boldsymbol{m}_t) \rangle + \frac{(L_0 + L_1\|\nabla F(\boldsymbol{x}_t)\|_2^q)\gamma^2}{2}\|\boldsymbol{u}_t\|_2^2 \\
=& F(\boldsymbol{x}_t) - \langle \nabla F(\boldsymbol{x}_t), \gamma \boldsymbol{u}_t \circ \text{Sign}(\nabla F(\boldsymbol{x}_t)) \rangle + \langle \nabla F(\boldsymbol{x}_t), \gamma \boldsymbol{u}_t \circ (\text{Sign}(\nabla F(\boldsymbol{x}_t)) - \text{Sign}(\boldsymbol{m}_t)) \rangle \\
& + \frac{\gamma^2(L_0 + L_1\|\nabla F(\boldsymbol{x}_t)\|_2^q)}{2}\|\boldsymbol{u}_t\|_2^2 \\
\leq& F(\boldsymbol{x}_t) - \langle \nabla F(\boldsymbol{x}_t), \gamma \boldsymbol{u}_t \circ \text{Sign}(\nabla F(\boldsymbol{x}_t)) \rangle + \gamma R\langle |\nabla F(\boldsymbol{x}_t)|, |\text{Sign}(\nabla F(\boldsymbol{x}_t)) - \text{Sign}(\boldsymbol{m}_t)| \rangle \\
& + \frac{\gamma^2 R^2 d(L_0 + L_1\|\nabla F(\boldsymbol{x}_t)\|_2^q)}{2},
\end{aligned} \tag{36}$$

where the last inequality holds due to $\boldsymbol{u}_t^{(j)} \leq \frac{1-\beta_1}{\sqrt{1-\beta_2}\sqrt{1-\frac{\beta_1^2}{\beta_2}}} = R, \forall j \in [d]$ according to Lemma 2.

Taking the expectation at the $t$-th iteration, we obtain

$$
\begin{aligned}
\mathbb{E}[F(\boldsymbol{x}_{t+1})] \leq & F(\boldsymbol{x}_t) - \gamma\langle\nabla F(\boldsymbol{x}_t), \mathbb{E}[\boldsymbol{u}_t] \circ \mathrm{Sign}(\nabla F(\boldsymbol{x}_t))\rangle + \gamma R\langle|\nabla F(\boldsymbol{x}_t)|, \mathbb{E}[|\mathrm{Sign}(\nabla F(\boldsymbol{x}_t)) - \mathrm{Sign}(\boldsymbol{m}_t)|]\rangle \\
& + \frac{\gamma^2 R^2 d(L_0 + L_1\|\nabla F(\boldsymbol{x}_t)\|_2^q)}{2} \\
\overset{(i)}{\leq} & F(\boldsymbol{x}_t) - \gamma\langle\nabla F(\boldsymbol{x}_t), \mathbb{E}[\boldsymbol{u}_t] \circ \mathrm{Sign}(\nabla F(\boldsymbol{x}_t))\rangle + 2\gamma R\,\mathbb{E}[\|\boldsymbol{m}_t - \nabla F(\boldsymbol{x}_t)\|_1] \\
& + \frac{\gamma^2 R^2 d(L_0 + L_1\|\nabla F(\boldsymbol{x}_t)\|_2^q)}{2} \\
\overset{(ii)}{\leq} & F(\boldsymbol{x}_t) - \gamma\langle\nabla F(\boldsymbol{x}_t), \mathbb{E}[\boldsymbol{u}_t] \circ \mathrm{Sign}(\nabla F(\boldsymbol{x}_t))\rangle + 2\gamma R\sqrt{d}\,\mathbb{E}[\|\boldsymbol{m}_t - \nabla F(\boldsymbol{x}_t)\|_2] \\
& + \frac{\gamma^2 R^2 d(L_0 + L_1\|\nabla F(\boldsymbol{x}_t)\|_2^q)}{2} \\
\overset{(iii)}{\leq} & F(\boldsymbol{x}_t) - \gamma\langle\nabla F(\boldsymbol{x}_t), \mathbb{E}[\boldsymbol{u}_t] \circ \mathrm{Sign}(\nabla F(\boldsymbol{x}_t))\rangle + 2\gamma R\sqrt{d}\,\mathbb{E}[\|\boldsymbol{m}_t - \nabla F(\boldsymbol{x}_t)\|_2] \\
& + \frac{\gamma^2 R^2 d(L_0 + L_1((1-q) + q\|\nabla F(\boldsymbol{x}_t)\|_2))}{2},
\end{aligned}
\tag{37}
$$

where $(i)$ holds due to $\mathbb{E}[|\mathrm{Sign}(\nabla F(\boldsymbol{x}_t^{(j)})) - \mathrm{Sign}(\boldsymbol{m}_t^{(j)})|] \leq 2\frac{\mathbb{E}[|\nabla F(\boldsymbol{x}_t^{(j)}) - \boldsymbol{m}_t^{(j)}|]}{|\nabla F(\boldsymbol{x}_t^{(j)})|} (\forall j \in [d])$ according to Lemma 3; $(ii)$ holds owing to the fact $\|\boldsymbol{a}\|_1 \leq \sqrt{d}\|\boldsymbol{a}\|_2$ for any $\boldsymbol{a} \in \mathbb{R}^d$; $(iii)$ holds due to the fact $a^q \leq (1-q) + qa$ according to Young's inequality.

Taking expectation from the $1$-st iteration to the $T$-th iteration and then summing them, we have

$$
\begin{aligned}
\mathbb{E}[F(\boldsymbol{x}_{t+1})] \leq & F(\boldsymbol{x}_1) - \gamma\sum_{t=1}^{T}\mathbb{E}[\|\boldsymbol{u}_t \circ \nabla F(\boldsymbol{x}_t)\|_1] + 2\gamma R\sqrt{d}\sum_{t=1}^{T}\mathbb{E}[\|\boldsymbol{m}_t - \nabla F(\boldsymbol{x}_t)\|_2] \\
& + \sum_{t=1}^{T}\frac{\gamma^2 R^2 d(L_0 + L_1((1-q) + q\mathbb{E}[\|\nabla F(\boldsymbol{x}_t)\|_2]))}{2}.
\end{aligned}
\tag{38}
$$

Rearranging the both sides and applying the facts that $F(\boldsymbol{x}_{t+1}) \geq F^*$, we obtain

$$
\begin{aligned}
& \frac{1}{T}\sum_{t=1}^{T}\mathbb{E}[\|\boldsymbol{u}_t \circ \nabla F(\boldsymbol{x}_t)\|_1] - \frac{\gamma R^2 q L_1 d}{2T}\sum_{t=1}^{T}\mathbb{E}[\|\nabla F(\boldsymbol{x}_t)\|_2] \\
\leq & \frac{F(\boldsymbol{x}_1) - F^*}{\gamma T} + \frac{2R\sqrt{d}}{T}\sum_{t=1}^{T}\mathbb{E}[\|\boldsymbol{m}_t - \nabla F(\boldsymbol{x}_t)\|_2] + \frac{\gamma R^2 d(L_0 + (1-q)L_1)}{2}.
\end{aligned}
\tag{39}
$$

Recalling $\boldsymbol{m}_t = \beta_1\boldsymbol{m}_{t-1} + (1-\beta_1)\boldsymbol{g}_t$, we obtain

$$
\begin{aligned}
\boldsymbol{m}_t - \nabla F(\boldsymbol{x}_t) = & (\beta_1\boldsymbol{m}_{t-1} + (1-\beta_1)\boldsymbol{g}_t) - \nabla F(\boldsymbol{x}_t) \\
= & \beta_1(\boldsymbol{m}_{t-1} - \nabla F(\boldsymbol{x}_{t-1})) + (1-\beta_1)(\boldsymbol{g}_t - \nabla F(\boldsymbol{x}_t)) - \beta_1(\nabla F(\boldsymbol{x}_t) - \nabla F(\boldsymbol{x}_{t-1})).
\end{aligned}
\tag{40}
$$

Utilizing recursion, we further have

$$\boldsymbol{m}_t - \nabla F(\boldsymbol{x}_t) = -\beta_1^t \nabla F(\boldsymbol{x}_1) + (1-\beta) \sum_{k=1}^{t} \beta_1^{t-k}(\boldsymbol{g}_k - \nabla F(\boldsymbol{x}_k)) - \sum_{k=1}^{t} \beta_1^{t-k+1}(\nabla F(\boldsymbol{x}_k) - \nabla F(\boldsymbol{x}_{k-1})),$$
(41)

where $\boldsymbol{m}_1 - \nabla F(\boldsymbol{x}_1) = -\beta_1 \nabla F(\boldsymbol{x}_1) + (1-\beta_1)(\boldsymbol{g}_1 - \nabla F(\boldsymbol{x}_1))$ due to $\boldsymbol{m}_0 = 0$.

Hence,

$$\frac{1}{T}\sum_{t=1}^{T}\mathbb{E}\left[\|\boldsymbol{m}_t - \nabla F(\boldsymbol{x}_t)\|_2\right] \leq \underbrace{\frac{1}{T}\sum_{t=1}^{T}\beta_1^t \|\nabla F(\boldsymbol{x}_1)\|_2}_{\mathcal{T}_1} + \underbrace{\frac{1-\beta_1}{T}\sum_{t=1}^{T}\mathbb{E}\left[\left\|\sum_{k=1}^{t}\beta_1^{t-k}\left(\boldsymbol{g}_k - \nabla F(\boldsymbol{x}_k)\right)\right\|_2\right]}_{\mathcal{T}_2}$$

$$+ \underbrace{\frac{1}{T}\sum_{t=1}^{T}\mathbb{E}\left[\left\|\sum_{k=1}^{t}\beta_1^{t-k+1}(\nabla F(\boldsymbol{x}_k) - \nabla F(\boldsymbol{x}_{k-1}))\right\|_2\right]}_{\mathcal{T}_3}$$
(42)

In terms of $\mathcal{T}_1$, we obtain

$$\mathcal{T}_1 = \frac{1}{T}\sum_{t=1}^{T}\beta_1^t \|\nabla F(\boldsymbol{x}_1)\|_2 \leq \frac{\|\nabla F(\boldsymbol{x}_1)\|_2}{T(1-\beta_1)}.$$
(43)

As for $\mathcal{T}_2$, we have

$$
\begin{aligned}
\mathcal{T}_2 =& \frac{1-\beta_1}{T} \sum_{t=1}^{T} \mathbb{E}\left[\left\|\sum_{k=1}^{t} \beta_1^{t-k}(\boldsymbol{g}_k - \nabla F(\boldsymbol{x}_k))\right\|_2\right] \\
&\overset{(i)}{\leq} \frac{1-\beta_1}{T} \sum_{t=1}^{T} \sqrt{\mathbb{E}\left[\left\|\sum_{k=1}^{t} \beta_1^{t-k}(\boldsymbol{g}_k - \nabla F(\boldsymbol{x}_k))\right\|_2^2\right]} \\
&\overset{(ii)}{=} \frac{1-\beta_1}{T} \sum_{t=1}^{T} \sqrt{\sum_{k=1}^{t} \beta_1^{2(t-k)} \mathbb{E}\left[\|\boldsymbol{g}_k - \nabla F(\boldsymbol{x}_k)\|_2^2\right]} \\
&\overset{(iii)}{\leq} \frac{1-\beta_1}{T} \sum_{t=1}^{T} \sqrt{\sum_{k=1}^{t} \beta_1^{2(t-k)}(\sigma_0^2 + \sigma_1^2 \mathbb{E}[\|\nabla F(x_k)\|_2^p])} \\
&\overset{(iv)}{\leq} \frac{1-\beta_1}{T} \sum_{t=1}^{T} \sqrt{\sum_{k=1}^{t} \beta_1^{2(t-k)}\sigma_0^2} + \frac{1-\beta_1}{T} \sum_{t=1}^{T} \sqrt{\sum_{k=1}^{t} \beta_1^{2(t-k)}\sigma_1^2 \mathbb{E}[\|\nabla F(x_k)\|_2^p]} \\
&\overset{(v)}{\leq} \frac{1-\beta_1}{T} \sum_{t=1}^{T} \sqrt{\sum_{k=1}^{t} \beta_1^{2(t-k)}\sigma_0^2} + \frac{1-\beta_1}{T} \sum_{t=1}^{T} \sqrt{\sum_{k=1}^{t} \beta_1^{2(t-k)}\sigma_1^2 \left(\frac{2-p}{2} + \frac{p}{2}\mathbb{E}[\|\nabla F(x_k)\|_2^2]\right)} \\
&\overset{(vi)}{\leq} \frac{1-\beta_1}{T} \sum_{t=1}^{T} \sqrt{\sum_{k=1}^{t} \beta_1^{2(t-k)}\sigma_0^2} + \frac{1-\beta_1}{T} \sum_{t=1}^{T} \sqrt{\sum_{k=1}^{t} \frac{(2-p)\sigma_1^2}{2} \beta_1^{2(t-k)}} \\
&\quad + \frac{1-\beta_1}{T} \sum_{t=1}^{T} \sqrt{\sum_{k=1}^{t} \frac{p}{2}\beta_1^{2(t-k)}\sigma_1^2 \mathbb{E}[\|\nabla F(x_k)\|_2^2]} \\
&\overset{(vii)}{\leq} \frac{1-\beta_1}{\sqrt{1-\beta_1^2}} \left(\sigma_0 + \sqrt{\frac{2-p}{2}}\sigma_1\right) + \frac{1-\beta_1}{T} \sum_{t=1}^{T} \sqrt{\sum_{k=1}^{t} \frac{p\sigma_1^2}{2} \beta_1^{2(t-k)} \mathbb{E}[\|\nabla F(x_k)\|_2^2]} \\
&\overset{(viii)}{\leq} \frac{1-\beta_1}{\sqrt{1-\beta_1^2}} \left(\sigma_0 + \sqrt{\frac{2-p}{2}}\sigma_1\right) + (1-\beta_1)\sqrt{\frac{1}{T}\sum_{t=1}^{T}\sum_{k=1}^{t} \frac{p\sigma_1^2}{2} \beta_1^{2(t-k)} \mathbb{E}[\|\nabla F(x_k)\|_2^2]} \\
&\overset{(ix)}{\leq} \frac{1-\beta_1}{\sqrt{1-\beta_1^2}} \left(\sigma_0 + \sqrt{\frac{2-p}{2}}\sigma_1\right) + (1-\beta_1)\sqrt{\frac{p\sigma_1^2}{2} \cdot \frac{1}{T}\sum_{t=1}^{T} \mathbb{E}[\|\nabla F(x_k)\|_2^2] \sum_{k=t}^{T} \beta_1^{2(t-k)}} \\
&\leq \frac{1-\beta_1}{\sqrt{1-\beta_1^2}} \left(\sigma_0 + \sqrt{\frac{2-p}{2}}\sigma_1\right) + \frac{\sqrt{p}\sigma_1(1-\beta_1)}{\sqrt{2(1-\beta_1^2)}} \sqrt{\frac{1}{T}\sum_{t=1}^{T} \mathbb{E}[\|\nabla F(x_k)\|_2^2]} \\
&\overset{(x)}{\leq} \frac{1-\beta_1}{\sqrt{1-\beta_1^2}} \left(\sigma_0 + \sqrt{\frac{2-p}{2}}\sigma_1\right) + \frac{C_0\sqrt{p}\sigma_1(1-\beta_1)}{\sqrt{2(1-\beta_1^2)}} \cdot \frac{1}{T}\sum_{t=1}^{T} \mathbb{E}[\|\nabla F(x_k)\|_2] \\
&\leq \sqrt{1-\beta_1} \left(\sigma_0 + \sqrt{\frac{2-p}{2}}\sigma_1\right) + C_0\sigma_1\sqrt{p(1-\beta_1)} \cdot \frac{1}{T}\sum_{t=1}^{T} \mathbb{E}[\|\nabla F(x_t)\|_2]
\end{aligned}
$$

(44)

where $(i)$ holds due to the fact $(\mathbb{E}[Z])^2 \leq \mathbb{E}[Z^2]$; $(ii)$ holds owing to $\mathbb{E}[\boldsymbol{g}_k - \nabla F(\boldsymbol{x}_k)] = \boldsymbol{0}$ according to Assumption C.3; $(iii)$ holds resulting from $\mathbb{E}\left[\|\boldsymbol{g}_k - \nabla F(\boldsymbol{x}_k)\|_2^2\right] \leq \sigma_0^2 + \sigma_1\|\nabla F(\boldsymbol{x}_k)\|_2^p$ according to Assumption C.3; $(iv)$ holds due to the fact $\sqrt{a+b} \leq \sqrt{a} + \sqrt{b}$; $(v)$ holds resulting from the fact that $a^p \leq \frac{2-p}{2} + \frac{p}{2}a^2, 0 \leq p \leq 2$ according to Young's inequality; $(vi)$ holds due to using the fact $\sqrt{a+b} \leq \sqrt{a} + \sqrt{b}$ again; $(vii)$ holds due to the fact $\sum_i^T a_i \leq \sqrt{T}\sqrt{\sum_i^T a_i^2}$ according to Cauchy-Schwaz inequality; $(viii)$ holds owing to $\sum_{k=1}^t \beta_1^{2(t-k)} \leq \frac{1}{1-\beta_1^2}$; $(ix)$ holds thanks to the fact $\sum_{i=1}^n \sum_{j=1}^i f(i,j)g(j) = \sum_{j=1}^n g(j) \sum_{i=j}^n f(i,j)$; $(x)$ holds because of Condition 1.

Now we turn attention to $\mathcal{T}_3$, *i.e.*,

$$
\begin{aligned}
\mathcal{T}_3 =& \frac{1}{T}\sum_{t=1}^T \mathbb{E}\left[\left\|\sum_{k=1}^t \beta_1^{t-k+1}(\nabla F(\boldsymbol{x}_k) - \nabla F(\boldsymbol{x}_{k-1}))\right\|_2\right] \\
\overset{(i)}{\leq}& \frac{1}{T}\sum_{t=1}^T\sum_{k=1}^t \beta_1^{t-k+1}\mathbb{E}\left[\|\nabla F(\boldsymbol{x}_k) - \nabla F(\boldsymbol{x}_{k-1})\|_2\right] \\
\overset{(ii)}{\leq}& \frac{1}{T}\sum_{t=1}^T\sum_{k=1}^t \beta_1^{t-k+1}\mathbb{E}\left[(L_0 + L_1\|\nabla F(\boldsymbol{x}_k)\|_2^q)\|\boldsymbol{x}_k - \boldsymbol{x}_{k-1}\|_2\right] \\
\overset{(iii)}{=}& \frac{1}{T}\sum_{t=1}^T\sum_{k=1}^t \beta_1^{t-k+1}\mathbb{E}\left[\gamma(L_0 + L_1\|\nabla F(\boldsymbol{x}_k)\|_2^q)\|\boldsymbol{u}_{t-1}\|_2\right]) \\
\overset{(iv)}{\leq}& \frac{1}{T}\sum_{t=1}^T L_0\gamma R\sqrt{d}\sum_{k=1}^t \beta_1^{t-k+1} + \frac{L_1\gamma R\sqrt{d}}{T}\sum_{k=1}^t \beta_1^{t-k+1}\mathbb{E}[\|\nabla F(\boldsymbol{x}_k\|_2^q)] \\
\leq& \frac{\gamma L_0 R\sqrt{d}}{1-\beta} + \frac{\gamma L_1 R\sqrt{d}}{T}\sum_{t=1}^T\sum_{k=1}^t \beta_1^{t-k+1}((1-q) + q\mathbb{E}[\|\nabla F(\boldsymbol{x}_k)\|_2]) \\
\overset{(v)}{\leq}& \frac{\gamma(L_0 + (1-q)L_1)R\sqrt{d}}{1-\beta_1} + \frac{\gamma q L_1 R\sqrt{d}}{T}\sum_{t=1}^T\sum_{k=1}^t \beta_1^{t-k+1}\mathbb{E}[\|\nabla F(\boldsymbol{x}_k)\|_2] \\
\overset{(vi)}{=}& \frac{\gamma(L_0 + (1-q)L_1)R\sqrt{d}}{1-\beta_1} + \frac{\gamma q L_1 R\sqrt{d}}{T}\sum_{k=1}^T \mathbb{E}[\|\nabla F(\boldsymbol{x}_k)\|_2]\sum_{t=k}^T \beta_1^{t-k+1} \\
\leq& \frac{\gamma(L_0 + (1-q)L_1)R\sqrt{d}}{1-\beta_1} + \frac{\gamma q L_1 R\sqrt{d}}{(1-\beta_1)T}\sum_{t=1}^T \mathbb{E}[\|\nabla F(\boldsymbol{x}_t)\|_2]
\end{aligned}
\tag{45}
$$

where $(i)$ holds due to the fact $\|\boldsymbol{a} + \boldsymbol{b}\|_2 \leq \|\boldsymbol{a}\|_2 + \|\boldsymbol{b}\|_2$; $(ii)$ holds owing to Assumption B.3; $(iii)$ holds due to the update rule; $(iv)$ holds depending on $\boldsymbol{u}^{(j)} \leq 1 - \beta_1/\sqrt{1-\beta_2}\sqrt{1 - \frac{\beta_1^2}{\beta_2}} = R$ according to Lemma 2; $(v)$ holds thanks to the fact that $a^q \leq (1-q) + qa$ according to Young's inequality; $(vi)$ holds resulting from the fact that $\sum_{i=1}^n \sum_{j=1}^i \boldsymbol{a}_{i,j} = \sum_{j=1}^n \sum_{i=j}^n \boldsymbol{a}_{i,j}$.

Combining Eq.(42) - Eq.(45), we have

$$\frac{1}{T}\sum_{t=1}^{T}\mathbb{E}\left[\|\boldsymbol{m}_t - \nabla F(\boldsymbol{x}_t)\|_2\right] \leq \frac{\|\nabla F(\boldsymbol{x}_1)\|_2}{T(1-\beta_1)} + \sqrt{1-\beta_1}\left(\sigma_0 + \sqrt{\frac{2-p}{2}}\sigma_1\right)$$

$$+ C_0\sigma_1\sqrt{p(1-\beta_1)} \cdot \frac{1}{T}\sum_{t=1}^{T}\mathbb{E}\left[\|\nabla F(x_t)\|_2\right] \tag{46}$$

$$+ \frac{\gamma R\sqrt{d}(L_0 + (1-q)L_1)}{1-\beta_1} + \frac{\gamma R\sqrt{d}qL_1}{1-\beta_1} \cdot \frac{1}{T}\sum_{t=1}^{T}\mathbb{E}[\|\nabla F(\boldsymbol{x}_t)\|_2]$$

Combining Eq.(39) and Eq.(46), we obtain

$$\frac{1}{T}\left(\sum_{t=1}^{T}\mathbb{E}[\|\boldsymbol{u}_t \circ \nabla F(\boldsymbol{x}_t)\|_1] - \left(\frac{\gamma R^2 dqL_1}{2} + 2C_0R\sqrt{d}\sigma_1\sqrt{p(1-\beta_1)} + \frac{2\gamma R^2 dqL_1}{1-\beta_1}\right)\sum_{t=1}^{T}\mathbb{E}[\|\nabla F(\boldsymbol{x}_t)\|_2]\right)$$

$$\leq \frac{F(\boldsymbol{x}_1) - F^*}{\gamma T} + \frac{2R\sqrt{d}\|\nabla F(\boldsymbol{x}_1)\|_2}{T(1-\beta_1)} + 2R\sqrt{d}\sqrt{1-\beta_1}\left(\sigma_0 + \sqrt{\frac{2-p}{2}}\sigma_1\right)$$

$$+ \frac{2\gamma R^2 d(L_0 + (1-q)L_1)}{1-\beta_1} + \frac{\gamma R^2 d(L_0 + (1-q)L_1)}{2}. \tag{47}$$

## B.4 Proof of Corollary 3

**Proof.** (1) Choosing $\bar{v} = \min_t \mathbb{E}[\boldsymbol{u}_t^{(j)}]$, we have

$$\sum_{t=1}^{T}\mathbb{E}[\|\boldsymbol{u}_t \circ \nabla F(\boldsymbol{x}_t)\|_1] = \sum_{t=1}^{T}\sum_{j=1}^{d}\mathbb{E}[|\boldsymbol{u}_t^{(j)}\nabla F(\boldsymbol{x}_t^{(j)})|]$$

$$\overset{(i)}{=} \sum_{t=1}^{T}\sum_{j=1}^{d}\mathbb{E}[\boldsymbol{u}_t^{(j)}]\mathbb{E}[|\nabla F(\boldsymbol{x}_t^{(j)})|]$$

$$\overset{(ii)}{=} \sum_{t=1}^{T}\mathbb{E}[\boldsymbol{u}_t^{(j)}]\sum_{j=1}^{d}\mathbb{E}[|\nabla F(\boldsymbol{x}_t^{(j)})|] \tag{48}$$

$$= \sum_{t=1}^{T}\mathbb{E}[\boldsymbol{u}_t^{(j)}]\mathbb{E}[\|(\nabla F(\boldsymbol{x}_t))\|_1]$$

$$\overset{(iii)}{\geq} \bar{v}\sum_{t=1}^{T}\mathbb{E}[\|(\nabla F(\boldsymbol{x}_t))\|_1]$$

$$\overset{(iv)}{=} \frac{\bar{v}\sqrt{d}}{C_1}\sum_{t=1}^{T}\mathbb{E}[\|(\nabla F(\boldsymbol{x}_t))\|_2],$$

where $(i)$ holds due to $\boldsymbol{u}_t^{(j)}$ and $|\nabla F(\boldsymbol{x}_t^j)|$ are mutually independent; $(ii)$ holds owing to applying Condition 2; $(iii)$ holds due to the condition $\bar{v} \leq \min_t \mathbb{E}[\boldsymbol{u}_t^{(j)}]$ ; $(iv)$ holds depending on Condition 3.

Then, we simplify the conclusion in Theorem 2 as

$$\left(\bar{v} - \left(\frac{\gamma C_1 R^2 \sqrt{d} q L_1}{2} + 2C_0 C_1 R \sigma_1 \sqrt{p(1-\beta_1)} + \frac{2\gamma C_1 R^2 \sqrt{d} q L_1}{1-\beta_1}\right)\right) \cdot \frac{1}{T} \sum_{t=1}^{T} \mathbb{E}[\|\nabla F(\boldsymbol{x}_t)\|_2]$$

$$\leq \frac{C_1(F(\boldsymbol{x}_1) - F^*)}{\gamma T \sqrt{d}} + \frac{2C_1 R \|\nabla F(\boldsymbol{x}_1)\|_2}{T(1-\beta_1)} + 2C_1 \sqrt{1-\beta_1} R\hat{\sigma} + \frac{2\gamma C_1 R^2 \sqrt{d} \hat{L}}{1-\beta_1} + \frac{\gamma C_1 R^2 \sqrt{d} \hat{L}}{2}. \tag{49}$$

Choosing $\gamma = \frac{C_2}{T^{3/4} d^{1/2}}$, $1 - \beta_1 = \frac{C_3}{T^{1/2}}$ and $T \geq (\frac{4C_1 C_2 R^2 q L_1}{C_3 \bar{v}} + \frac{4C_0 C_1 \sqrt{C_3} R\sigma_1 \sqrt{p}}{\bar{v}} + (\frac{C_1 C_2 R^2 q L_1}{\bar{v}})^{1/3})^4$, following Lemma 4, it holds that

$$\frac{\gamma C_1 R^2 \sqrt{d} q L_1}{2} + 2C_0 C_1 R\sigma_1 \sqrt{p(1-\beta_1)} + \frac{2\gamma C_1 R^2 \sqrt{d} q L_1}{1-\beta_1} \leq \frac{\bar{v}}{2}. \tag{50}$$

Then, we reformulate Eq. (49) as

$$\frac{1}{T} \sum_{t=1}^{T} \mathbb{E}[\|\nabla F(\boldsymbol{x}_t)\|_2] \leq \frac{C_1}{\bar{v}} \left(\frac{2(F(\boldsymbol{x}_1) - F(\boldsymbol{x}^*))}{C_2 T^{1/4}} + \frac{4R \|\nabla F(\boldsymbol{x}_1)\|_2}{C_3 T^{1/2}} + \frac{4C_3 R\hat{\sigma}}{T^{1/4}} + \frac{4C_2 R^2 \hat{L}}{C_3 T^{1/4}} + \frac{C_2 R^2 \hat{L}}{T^{3/4}}\right). \tag{51}$$

(2) Using generalized Young's inequality, we minimize the bottle-neck terms to obtain the lowerest bound on the right hand of Eq. (49), *i.e.*,

$$\frac{C_1(F(\boldsymbol{x}_1) - F^*)}{\gamma T \sqrt{d}} + 2C_1 \sqrt{1-\beta_1} R\hat{\sigma} + \frac{2C_1 \gamma R^2 \sqrt{d} \hat{L}}{1-\beta_1}$$

$$\geq \left(\frac{4C_1(F(\boldsymbol{x}_1) - F^*)}{\gamma T \sqrt{d}}\right)^{1/4} \cdot \left(4C_1 \sqrt{1-\beta_1} R\hat{\sigma}\right)^{1/2} \cdot \left(\frac{8C_1 \gamma R^2 \sqrt{d} \hat{L}}{1-\beta_1}\right)^{1/4} \tag{52}$$

$$= \frac{512^{1/4} C_1 R\hat{\sigma}^{1/2} \hat{L}^{1/4}(F(\boldsymbol{x}_1) - F^*)^{1/4}}{T^{1/4}},$$

where the lowest bound achieved if and only if $\frac{4C_1(F(\boldsymbol{x}_1)-F^*)}{\gamma T \sqrt{d}} = 4C_1 \sqrt{1-\beta_1} R\hat{\sigma}$ and $4C_1 \sqrt{1-\beta_1} R\hat{\sigma} = \frac{8C_1 \gamma R^2 \sqrt{d} \hat{L}}{1-\beta_1}$, and we further obtain

$$\gamma = \frac{(F(\boldsymbol{x}_1) - F^*)^{3/4}}{2^{1/4} T^{3/4} d^{1/2} R\hat{\sigma}^{1/2} \hat{L}^{1/4}},$$

$$\beta_1 = 1 - \frac{2^{1/2} \hat{L}^{1/2}(F(\boldsymbol{x}_1) - F^*)^{1/2}}{T^{1/2} \hat{\sigma}}. \tag{53}$$

When it is chosen $T \geq (\frac{4C_1 \hat{C}_2 R^2 q L_1}{\hat{C}_3 \bar{v}} + \frac{4C_0 C_1 \sqrt{\hat{C}_3} R\sigma_1 \sqrt{p}}{\bar{v}} + (\frac{C_1 \hat{C}_2 R^2 q L_1}{\bar{v}})^{1/3})^4$ where $\hat{C}_2 = \frac{(F(\boldsymbol{x}_1)-F^*)^{3/4}}{2^{1/4} R\hat{\sigma}^{1/2} \hat{L}^{1/4}}$ and $C_3 = \frac{2^{1/2} \hat{L}^{1/2}(F(\boldsymbol{x}_1)-F^*)^{1/2}}{\hat{\sigma}}$, following Lemma 4, it holds that

$$\frac{\gamma C_1 R^2 \sqrt{d} q L_1}{2} + 2C_0 C_1 R\sigma_1 \sqrt{p(1-\beta_1)} + \frac{2\gamma C_1 R^2 \sqrt{d} q L_1}{1-\beta_1} \leq \frac{v}{2}. \tag{54}$$

Then, we reformulate Eq. (49) as

$$\frac{1}{T}\sum_{t=1}^{T}\mathbb{E}[\|\nabla F(\boldsymbol{x}_t)\|_2] \leq \frac{C_1}{\bar{v}}\left(\frac{512^{1/4}R\hat{\sigma}^{1/2}\hat{L}^{1/4}(F(\boldsymbol{x}_1)-F^*)^{1/4}}{T^{1/4}} + \frac{4R\|\nabla F(\boldsymbol{x}_1)\|_2}{\hat{C}_3 T^{1/2}} + \frac{\hat{C}_2 R^2 \hat{L}}{T^{3/4}}\right).$$

(55)

### B.5 Convergence of Adam without Condition 1-3

In the proof of Lemma 2, Condition 1 is no longer necessary when Assumption C.1 is used. Even in the absence of Conditions 1, 2, and 3, and assuming no access to the oracle values $L_0$, $L_1$, $\sigma_0$, and $\sigma_1$, we can still theoretically establish that $\frac{1}{T}\sum_{t=0}^{T-1}\mathbb{E}[\|\nabla F(\boldsymbol{x}_t)\|_2]$ for Adam converges at the rate of $O\left(\frac{1}{T^{1/4}}\right)$, under the assumptions of $(L_0, L_1, q)$-smoothness and affine variance.

**Theorem 4 (Convergence of Adam without Condition 1-3)** *Let* $\{x_t\}_{t=0}^{T-1}$ *be generated by Algorithm 1. Suppose that Assumptions A, B.3 and C.1 hold. Define* $u_t^{(j)} := |\boldsymbol{m}_t^{(j)}|/(\sqrt{\boldsymbol{v}_t^{(j)}} + \epsilon)$, $R := 1 - \beta_1/\sqrt{(1-\beta_2)(1-\beta_1^2/\beta_2)}$ *and* $\hat{L} := L_0 + (1-q)L_1$. *Choose* $\gamma = \frac{C_2}{T^{3/4}d^{1/2}}$, $\beta_1 < \sqrt{\beta_2}$, $1-\beta_1 = \frac{C_3}{T^{1/2}}$ *and* $0 < v \leq \min_{t,j} u_t^{(j)}$. *Then, it holds for any* $T \in \mathbb{N}^+$ *and* $T \geq (\frac{4C_2 R^2 d^{1/2}qL_1}{C_3 v} + (\frac{C_2 R^2 d^{1/2}qL_1}{v})^{1/3})^4$,

$$\frac{1}{T}\sum_{t=0}^{T-1}\mathbb{E}[\|\nabla F(\boldsymbol{x}_t)\|_1] \leq \frac{1}{v}\left(\frac{2(F(\boldsymbol{x}_0)-F(\boldsymbol{x}^*))}{C_2 T^{1/4}d^{1/2}} + \frac{4Rd^{1/2}\|\nabla F(\boldsymbol{x}_0)\|_2}{C_3 T^{1/2}}\right.$$
$$\left. + \frac{4C_3 Rd^{1/2}\sigma_0}{T^{1/4}} + \frac{4C_2 R^2 d^{1/2}\hat{L}}{C_3 T^{1/4}} + \frac{C_2 R^2 d^{1/2}\hat{L}}{T^{7/4}}\right).$$

(56)

**Proof.** When C.1 holds, it implies $\sigma_1 = 0$. Also, it is known that $\|\nabla F(\boldsymbol{x}_t)\|_2 \leq \|\nabla F(\boldsymbol{x}_t)\|_1$. Hence, the conclusion in Theorem 2 can be simplified as

$$\left(v - \frac{\gamma R^2 dqL_1}{2} - \frac{2\gamma R^2 dqL_1}{1-\beta_1}\right)\cdot\frac{1}{T}\sum_{t=1}^{T}\mathbb{E}[\|\nabla F(\boldsymbol{x}_t)\|_1] \leq \frac{F(\boldsymbol{x}_1)-F^*}{\gamma T} + \frac{2R\sqrt{d}\|\nabla F(\boldsymbol{x}_1)\|_2}{T(1-\beta_1)}$$
$$+ 2\sqrt{1-\beta_1}R\sqrt{d}\sigma_0 + \frac{2\gamma R^2 d\hat{L}}{1-\beta_1} + \frac{\gamma R^2 d\hat{L}}{2},$$

(57)

where $0 < v \leq \min_{t,j}\boldsymbol{u}_t^{(j)}$ and $\hat{L} = L_0 + (1-q)L_1$.

Choosing $\gamma = \frac{C_2}{T^{3/4}d^{1/2}}$ and $1-\beta_1 = \frac{C_3}{T^{1/2}}$ and $T \geq (\frac{4C_2 R^2 d^{1/2}qL_1}{C_3 v} + (\frac{C_2 R^2 d^{1/2}qL_1}{v})^{1/3})^4$, following Lemma 4, it holds that

$$\frac{\gamma R^2 dqL_1}{2} + \frac{2\gamma R^2 dqL_1}{1-\beta_1} \leq \frac{v}{2}.$$

(58)

Then, we arrive the conclusion

$$\frac{1}{T}\sum_{t=1}^{T}\mathbb{E}[\|\nabla F(\boldsymbol{x}_t)\|_1] \leq \frac{1}{v}\left(\frac{2d^{1/2}(F(\boldsymbol{x}_1)-F(\boldsymbol{x}^*))}{C_2 T^{1/4}} + \frac{4Rd^{1/2}\|\nabla F(\boldsymbol{x}_1)\|_2}{C_3 T^{1/2}}\right.$$
$$\left. + \frac{4C_3 Rd^{1/2}\sigma_0}{T^{1/4}} + \frac{4C_2 R^2 d^{1/2}\hat{L}}{C_3 T^{1/4}} + \frac{C_2 R^2 d^{1/2}\hat{L}}{T^{3/4}}\right).$$

(59)