# OpenReview forum: "Simple Convergence Proof of Adam From a Sign-like Descent Perspective"
_ICLR.cc/2026/Conference — Submitted to ICLR 2026_

### Official Review · Reviewer_qGsD · 2025-10-19

**Soundness:** 2
**Presentation:** 3
**Contribution:** 2
**Rating:** 2
**Confidence:** 5

**Summary:**

This paper provides a convergence proof for Adam from the perspective of sign-like descent, which simplifies the convergence analysis under generalized-smooth nonconvex optimization.

**Strengths:**

1. Although it is widely believed that Adam is close to sign descent, this paper provides a set of conditions to ensure when such a perspective is valid (Condition 1, 2, 3, decreasing rate of gradient norm over time in Proposition 1).

2. The presentation of this paper is good, with a comprehensive comparison with prior work.

**Weaknesses:**

1. Some of the these extra conditions the authors assume are not reasonable. For example, the authors assume that the gradient norm is decreasing over time with certain rate in Proposition 1 to argue that Condition 1 commonly holds in practice. However, the gradient norm may not decrease over time in practice and in theory. For example, even the averaged gradient decrease over time, it does not indicate that the gradient itself is monotonically decreasing.

2. Despite the numerical experiments of hypothesis testing, I do not think Condition 2 is reasonable. Adam’s coordinate updates within each layer are neither independent nor identically distributed. In practice, all weight gradients in a layer share common input activations and backpropagated errors, which induces strong correlations across coordinates. In addition, Condition 2 assumes i.i.d. condition for **every coordinate**, not just within a single layer. This is unrealistic, as coordinates from adjacent layers are inherently statistically dependent due to the recursive structure of backpropagation.

**Questions:**

1. Can you justify the extra conditions?

---

### Official Review · Reviewer_3zA9 · 2025-10-22

**Soundness:** 1
**Presentation:** 3
**Contribution:** 2
**Rating:** 2
**Confidence:** 4

**Summary:**

This paper presents a convergence analysis of Adam for (generalized) smooth, non-convex functions with (generalized) affine variance, assuming some additional conditions. The analysis works by considering Adam as a kind of adaptive sign descent, rather than as a preconditioned gradient descent. Some experimental support for the additional conditions is also provided.

**Strengths:**

1. The problem considered is important. The convergence of Adam is a long-standing area of research that is very relevant to deep learning in general.
2. The paper is well-written and clear about its stated contributions.

**Weaknesses:**

1. The reliance on the additional conditions (Conditions 1-3) is a significant weakness for several reasons. At best these conditions obscure the results, and at worst the conditions are simply not valid.

    a. I did not find any convincing evidence for Condition 1, as numerical validation is only provided for Conditions 2 and 3. This is odd considering that Condition 1 seems like it would be quite simple to verify/disprove numerically. Proposition 1 is intended to provide theoretical support for Condition 1, but Proposition 1 depends on the assumption that $\|\nabla F(x_t)\|_2 = \Theta(t^{-\alpha})$. Not only does this assumption assume convergence to an approximate stationary point (which the paper aims to prove), but it also assumes a lower bound on this rate of convergence, for which I see no evidence at all. The discussion on Lines 221-224 is not convincing to me.

    b. From Condition 2, I agree that the variable $\bar{v}$ (defined as the lower bound of $\bar{u}_t$) should be larger than $0$, but a subtle issue is that $\bar{v}$ can depend on $d$ or $T$. For example, what if for some particular $F$ it holds that $\bar{v} = 1/d$ or $\bar{v} = 1/T$? Since the convergence rates (e.g. Equations 10 and 11) have a coefficient of $1/\bar{v}$ in front, this means that the rate would be $d/T^{1/4}$ when $\bar{v} = 1/d$ or $T^{3/4}$ when $\bar{v} = 1/T$! So it is not even clear to me that the convergence rates are non-vacuous. This is why I previously said that the conditions obscure the results.

    c. As stated, Condition 3 simply cannot be true. Condition 3 assumes that the ratio $\|\nabla F(x_t)\|_1 / \|\nabla F(x_t\|_2$ is constant across iterations $t$, which is definitely not true. Perhaps this condition can be weakened to assume that this ratio is upper and lower bounded by two different constants, but this would require significant modifications to the proof. To be clear: I am not requesting that the authors modify their proof to address this during the rebuttal window, as this seems like a tedious task that may not be worth the effort.

    d. Theorem 4 is supposed to show convergence without assuming Conditions 1-3, but it seems that Theorem 4 still uses Condition 2 through the variable $v$. In any case, the rate from Theorem 4 depends on $d^{1/2}$, which is a significantly weaker result than the main result.

2. There are several instances of unclear or exaggerated language which I think should be clarified. First, the paper is framed as relying on weaker assumptions than previous work, but as I pointed out in my Weakness #1, this paper relies on several dubious conditions not used in prior work. Second, it is claimed several times that the "sign descent" perspective simplifies the proof process, but it isn't clear to me that this is the case: the proof in this paper looks comparably long and complicated as previous analyses of Adam. Third, it is claimed several times throughout the paper that the interpretation of Adam as an adaptive sign descent is a novel perspective, e.g. in the conclusion: "This work breaks with convention and provides a pioneering reinterpretation of Adam as a sign-like descent algorithm". I think this is an inappropriate exaggeration, given that Balles & Hennig (2018) (and probably several other works since) have already established this perspective, even if they didn't use this perspective to develop new convergence proofs. Calibrating the tone of these claims will strengthen the paper.

Typos/small suggestions:
- As stated, the second part of Condition 2 ($\bar{u}_t > 0$) is trivially true. It seems like you may instead want $\bar{u}_t \geq \bar{v} > 0$.
- Caption of Table 1 refers to Theorem B.6. I think you mean Theorem 4.

**Questions:**

1. Do you have any numerical evidence to support Condition 1?
2. Numerically, do you find that $\bar{v}$ from Condition 2 depends on $d$ or $T$?
3. Am I correct in understanding that Theorem 4 still depends on Condition 2?

---

### Official Review · Reviewer_6JTc · 2025-10-26

**Soundness:** 2
**Presentation:** 3
**Contribution:** 2
**Rating:** 2
**Confidence:** 4

**Summary:**

This paper provides a convergence proof for Adam from the perspective of sign-like descent instead of the commonly used preconditioned stochastic gradient descent with momentum. Based on this reformulation, they are able to prove the optimal rate of $O(T^{-1/4})$ for Adam under
the generalized $p$-affine variance and $(L\_0, L\_1, q)$-smoothness, without dependence on the model dimensionality or the numerical stability parameter.

**Strengths:**

(a). The major contribution in this paper lies in the theoretical part, where the convergence rate for Adam is established under the general weak assumptions, which are more realistic to reflect the practical training.

(b). The convergence rate is tight, and the result is novel to my best knowledge, in the context of general weak assumptions.

(c). The treatment of Adam as a sign-like gradient descent, although not proposed at the first time, is still novel in the convergence analysis of Adam in non-convex stochastic optimization.

**Weaknesses:**

My major concerns also lie in the theoretical part.

It's clear to see that the convergence rate is established based not only on the weak assumptions such as generalized smoothness and affine noise, but also heavily on Conditions 1-3. These conditions, however, do not sound so natural and reasonable. Specifically,
- Condition 1: Although the authors propose Proposition 1 to illustrate this condition, the assumption $\\|\nabla F(x\_t) \\|\_2$ decreases with $O(1/t^{\alpha})$ is not so reasonable. Existing results, such as Arjevani et al. (2023), only provide the guarantee of $O(1/T^{1/4})$ for $\min_{t \in [T]} \\|\nabla F(x\_t) \\|\_2$ or $\sum\_{t=1}^T \\|\nabla F(x\_t) \\|\_2/T$ instead of the precise description on the decreasing rate. Moreover, in the practical training, $\\|\nabla F(x\_t) \\|\_2$ usually does not follow such a decreasing formula.
- Condition 2 (I): The i.i.d. assumption on each coordinate of the update is not convincing. The authors provide a two-sample K-S test to illustrate this assumption. However, to my best knowledge, the K-S test is used to verify whether two distributions are the same or not. Also, in practical training, the i.i.d. is usually satisfied.

- Condition 2 (II): the lower bound assumption of update is one of the central points in the convergence result, showing in the term $\bar{v} = \min_{t \in [T]} E[u\_t^{(j)}]$. First, the lower bound is not commonly used in literature, perhaps only in (Guo et al., 2020). This approach sidesteps the key difficulty in analyzing Adam, such as the correlation of adaptive stepsize and stochastic gradient. Second, even though the lower bound exists, the specific value of $\bar{v}$ remains unknown. It's not convincing to treat it as a constant without any further theoretical or empirical evidence.

**Questions:**

There are some confusing notations in this paper, which deserve a more careful check.

- The notation $\mathbb{E}$ is confusing. For example, in Eq. (6) and (7), the expectation should be the conditional expectation given $x$, and in Eq. (12) and (13), it's the conditional expectation given $x_t$. However, $\mathbb{E}$ in convergence results refers to the total expectation.

- the notations $u_{t}^{(j)}$ and ${\bf u}_t^{(j)}$ are confused. Usually, the scalar terms use the light notation while the vector or matrix terms use the bold notation.

---

### Official Review · Reviewer_PwXu · 2025-10-29

**Soundness:** 2
**Presentation:** 1
**Contribution:** 1
**Rating:** 2
**Confidence:** 3

**Summary:**

The paper recasts Adam as a sign-like descent algorithm by grouping the update magnitude $u_t = m_t / (\sqrt{v_t} + \epsilon)$ and proving an $ O(1/\sqrt{T})$ non-convex convergence rate that, according to the authors, is (i) dimension-free, (ii) independent of the stabilising constant $\epsilon$, and (iii) obtained under generalized smoothness assumptions.

The derived results also need some extra new introduced conditions.

The main results are Theorem 2 and Corollary 3.

The key assumptions include:

- Generalized ( p )-affine variance of stochastic gradients,
- ((L_0, L_1, q))-smoothness of the objective function,
- Conditions 1–3, and
- A lower bound condition on the normalized momentum:
$ \mathbb{E}\frac{|m_t^j|}{\sqrt{v_t^j + \epsilon}} \geq \bar{\nu} > 0. $

Under these assumptions, the derived results establish upper bounds on different convergence measures, including:

- Theorem 3 (without requiring Conditions 2–3): $ {1\over T} \left( \sum_{t=1}^{T-1} E |u_t \nabla F(x_t)|_1-c\sum_{t=1}^{T-1} E |\nabla F(x_t)|_2 \right).$
- Corollary 3: $ \frac{1}{T} \sum_{t=1}^{T-1} \mathbb{E} |\nabla F(x_t)|_2. $

**Strengths:**

- Studies Adam convergence from **sign-gradient-based algorithms** perspective
- Builds on motivation from Balles and Hennig (2018) and related works, as noted on Page 3.

**Weaknesses:**

- The measure ${1\over T} \sum_{t=1}^{T-1} \mathbb{E}|u_t \nabla F(x_t)|_1 $ in Theorem 2 could be meaningless, as $u_t^j$ in some cases could be very small.

- Conditions 1-3 and the lower bound condition $E |m_t^j| /\sqrt{v_t^j + \epsilon} \geq \bar{v} >0$ are unnatural and strong. All existing convergence analysis for Adam in the nonconvex smooth optimization do not involve these assumptions.

- The comparisons with state-of-the-art results in Table 1 are unfair because the authors do not account for the new additional assumptions (Conditions 1-3 and the lower bound condition $E |m_t^j| /\sqrt{v_t^j + \epsilon} \geq \bar{v} >0$) introduced in this paper.

- The upper bound for Adam in Corollary 3 depends on ${1 \over \bar{v}}$, where $E |m_t^j| /\sqrt{v_t^j + \epsilon} \geq \bar{v} >0$, which could be very large in some cases.

- The numerical results for supporting Conditions 1-3 and the lower bound condition are limited. And the constant setup on $\beta_1$ and $\beta_2$ in numerical results seems not to be consistent with the theoretical results on $\beta_1$ (or $\beta_2$) which are depending on the total number of iterations.

- "$|\nabla F(x_t)|_2$ decreases at the rate $1/t^{\alpha}$" from  Proposition 1 is wrongly stated. According to the proof in Appendix, the authors essentially mean that $c/t^{\alpha} \leq |\nabla F(x_t)|_2  \leq C /t^{\alpha}$.

- Proposition 1 is insufficient to support the theoretical validity of Condition 1. The authors' logic appears circular: they use Condition 1 to prove that $|\nabla F(x_t)|_2  \leq C /t^{\alpha}$, and then justify the reasonableness of Condition 1 based on the resulting bound $c/t^{\alpha} \leq |\nabla F(x_t)|_2  \leq C /t^{\alpha}$.

- Some of the statement are wrongly stated or over-claimed. Take a few as example as follows.
  - Abstract, the mild conditions in the statement `For the first time, with some mild conditions, we prove that Adam achieves' are not truly mild.
  - Abstract, `Additionally, our theoretical analysis provides new insights into the role of momentum as a key factor ensuring convergence'. `The role of momentum as a key factor ensuring convergence' is now a relatively  well known result. See e.g. (Xie S, Mohamadi M A, Li Z. Adam Exploits $\ell_\infty $-geometry of Loss Landscape via Coordinate-wise Adaptivity[J]. arXiv preprint arXiv:2410.08198, 2024; Li H, Dong Y, Lin Z. On the $ O (\frac {\sqrt {d}}{T^{1/4}}) $ Convergence Rate of RMSProp and Its Momentum Extension Measured by $\ell_1 $ Norm[J]. arXiv preprint arXiv:2402.00389, 2024; ) and the earlier related papers.  In fact, (15) is similar to equations found in some of the literature, but the authors do not mention this."
  - Line 101, `Recently, Muon Jordan et al. (2024), an extended matrix-sign optimizer'. The statement matrix-sign is not precise.
  - Lines 633-634,  'Shi et al. (2021) and Zhang et al. (2022) respectively proved random-shuffled AMSProp and Adam will
converge to the neighbourhood of stationary points with the rate'. The latter  can also derive similar convergence rates if one chooses $\beta_1$ and $\beta_2$ that are depending on $T$.
  - Lines 658-659, `Eq. (41), Eq. (56) and Eq. (58) in the proof suggest that...'.  The authors fail to note that $\eta = \sqrt{1-\beta_2}= 1/\sqrt{T}$ in that paper. Thus, the bounds (41), (56), and (58) (which are also related to $\eta$) are $\mathcal{O}(\log T)$.
  - Please revise Page 20 to use space more efficiently. We also identified similar issues with excessive spacing in other sections.
  - The other places...

The manuscript appears to be underprepared; it requires thorough proofreading, revision, and a further round of review.

**Questions:**

- Could Conditions 1-3 and the lower bound condition be further validated through more comprehensive experiments?

 - See the weakness part.

---

### Meta-Review · Area_Chair_1LK6 · 2025-12-20

**Summary:**

The main contribution of the paper is a convergence analysis of Adam that reframes it as a sign-like descent method, claiming dimension-free and $\epsilon$-independent rates under generalized smoothness and affine variance assumptions.

While a new perspective for the analysis of Adam is welcome, the reviewers have identified several important shortcomings such as:
1) Reliance on additional Conditions 1–3 and a lower bound on normalized momentum, which reviewers widely consider as strong and unrealistic assumptions.
2) Fruthermore, several convergence measures (e.g., in Theorem 2 and Corollary 3) may be vacuous or meaningless because key quantities can be arbitrarily small or large, making the bounds uninformative.
3) Some claims are viewed as overstated, particularly regarding the novelty of the sign-descent interpretation and the role of momentum, which are already known in the literature.
Numerical experiments are considered insufficient and inconsistent with theory, providing weak or incomplete validation of the assumptions.

Overall, the manuscript is seen as not being ready for publication and requires substantial revision. Also due to the lack of responses from the authors, I'm unable to recommend acceptance.

**Reviewer Concerns:**

No rebuttal provided.

**Reviewer Scores:**

N/A

---

### Decision · Program_Chairs · 2026-01-26

Reject